# Powder-size driven facile microstructure control in powder-fusion metal additive manufacturing processes

Shubham Chandra[1], Chengcheng Wang[1], Shu Beng Tor [2],
Upadrasta Ramamurty [2,3] & Xipeng Tan [4,5] ✉

Microstructure control in metal additive manufacturing is highly desirable for superior and bespoke mechanical performance. Engineering the columnar-to-equiaxed transition during rapid solidification in the additive manufacturing process is crucial for its technological advancement. Here, we report a powder-size driven melt pool engineering approach, demonstrating facile and large-scale control in the grain morphology by triggering a counterintuitive response of powder size to the additively manufactured 316 L stainless steel microstructure. We obtain coarse-grained (>100 µm) or near-monocrystalline microstructure using fine powders and near-equiaxed, fine-grained (<10 µm) microstructure using coarse powders. This approach shows resourceful adaptability to directed energy deposition and powder-bed fusion with no added cost, where the particle-size dependent powder-flow preheating effects and powder-bed thermophysical properties drive the microstructural variations. This work presents a pathway for leveraging feedstock particle size distribution towards more controllable, cost-effective, and sustainable metal additive manufacturing.

Producing metallic parts with properties superior to their conventional counterparts has emerged as the primary focus of research in metal additive manufacturing (AM). The mechanical properties of an additively manufactured component, free from processing-induced defects, depend largely on its microstructure. Hence, the ability to control microstructural evolution during part fabrication can be beneficial in producing complex shapes with predictable mechanical properties. Moreover, the ability to tailor the microstructure should be adaptable to various AM techniques. This is particularly demanded for stainless steel 316 L (SS316L) due to its widespread applications for general purposes in many industry sectors under harsh and corrosive environments[1,2].

Most of the research reporting attractive mechanical properties through AM has been focused on grain refinement or their columnar-to-equiaxed transitions (CET). A fine-grained equiaxed microstructure is desirable in AM as it is resistant to hot cracks and grants a fairly isotropic mechanical response[3,4]. This in turn undermines the growth of weaker, crack-prone, and directional property-inducing long columnar grains[5–8]. Though the growth of columnar grains might encompass several limitations, it is indeed the first step towards producing microstructure with a superior creep life[9] and, eventually, a single-crystal microstructure[10,11]. To date, the most effective ways to achieve CET in a variety of powder fusion-additive manufacturing (PF-AM) processes include variation of processing conditions[3,12–14], laser

[1]Singapore Centre for 3D Printing, School of Mechanical and Aerospace Engineering, Nanyang Technological University, 50 Nanyang Avenue, Singapore 639798, Singapore. [2]School of Mechanical and Aerospace Engineering, Nanyang Technological University, 50 Nanyang Avenue, Singapore 639798, Singapore. [3]Institute of Materials Research and Engineering, Agency for Science, Technology and Research, 138634 Singapore, Singapore. [4]Department of Mechanical Engineering, National University of Singapore, 9 Engineering Drive 1, Singapore 117575, Singapore. [5]Department of Materials Science and Engineering, National University of Singapore, 9 Engineering Drive 1, Singapore 117575, Singapore. ✉e-mail: xptan@nus.edu.sg

beam shaping[15], alloy composition redesign[16], nanometer scale (nanoscale) inclusion induced heterogeneous nucleation[16,17], high-intensity ultrasound triggered grain refinement[18] or post-processing driven recrystallization[19]. Among these, the last four have been reported so far in a capacity of unidirectional microstructural evolution control, allowing transitions from columnar-to-equiaxed and not vice-versa. Also, all of these require additional experimental efforts, costs, and lack the adaptability to the directed energy deposition (DED) and powder-bed fusion (PBF) PF-AM processes alike. Hence, it is necessary to explore viable alternatives that are economical yet adaptable to achieve either an equiaxed, fine-grained (FG) microstructure or a coarse-grained near-monocrystalline microstructure in the as-built condition.

We hypothesize that the sizes and velocities of distinct particles impinging the melt pool in a DED process and the collective thermophysical properties of the powder bed in a PBF process should impact the melt pool geometry—hence its solidification outcome. This opens a venue for microstructural variation not explored to date. For PBF processes, it has been established that the powder-bed apparent density is improved for a large particle size distribution (PSD)[20–22]. Additionally, measurements of the rheological properties of powders show higher flowability and spreadability for wide PSDs, particularly benefitting from coarse powder particles[23,24]. For the PBF processes in general, the thermophysical properties of a bed are supposed to impact the melt pool heat dissipation, its geometry and solidification conditions[25,26]. However, the research exploring the effect of PSD on the as-built microstructure and the mechanical properties has been primarily limited to laser powder bed fusion (L-PBF) processes where the typical PSD range is 15–63 μm. This has led to a constrained understanding that fine powders generate fine microstructure and/or superior mechanical properties[20,27,28], with certain exceptions[24]. We believe that these observations might be valid for a limited range of PSDs but cannot be generalized to correlate the powder size to the expected microstructure/mechanical properties as evident from the exceptions observed in the consensus[24]. This becomes increasingly crucial for electron beam powder bed fusion (E-PBF) and laser directed energy deposition (L-DED) processes and in the recently developed high-power L-PBF processes[29] where broader PSDs can be utilized. Moreover, it offers the possibility to capitalize on the latter to produce customizable microstructures.

In this work, we explore the influence of PSD on the microstructural evolution for SS316L printed using the L-DED process by systematically varying powder sizes for a constant set of processing parameters. This controlled investigation from the deposition of single tracks to block samples helps in isolating the response of microstructural evolution to variations in PSD. Leveraging this role, we report a melt pool engineering (MPE) approach where we demonstrate site-specific microstructure control using L-DED. Additionally, we leverage the impact of varying PSDs on powder-bed thermophysical properties, and consequently on the melt pool solidification behavior, to achieve a bi-directional control of microstructural evolution in the E-PBF process. This approach results in a fine-equiaxed and a coarse-columnar microstructure for coarse and fine PSDs, respectively, with no added cost.

## Results

### Feedstock particle size-preheat temperature correlation in L-DED process

In the L-DED process, the powder particles heat up as they traverse through the zone of laser irradiance before striking the melt pool[30,31]. Assuming a negligible thermal gradient within a particle and ignoring the particle-driven laser shielding, the rise in a particle's temperature due to the irradiance by a laser[30] can be calculated by the following relation:

$$\Delta T = \left( \frac{3\alpha_p P}{\pi r_b^2 \rho_p C_p} \right) \cdot \frac{t_f}{D_p} \tag{1}$$

where $\Delta T$ is the rise in temperature or particle preheat temperature, $D_p$ is the particle diameter, $\rho_p$ is particle density, $Cp$ is the specific heat capacity of the powder material, $\alpha_p$ is laser absorptivity, $P$ is the laser power, $r_b$ is the laser spot radius, and $t_f$ is the time of flight of the particle through the illumination zone (details in Supplementary Note 4).

It is evident from Eq. (1) that the in-flight temperature rise of particles in a feedstock for the same process parameters and same material properties will depend upon their size and time of flight through the zone. The effect of increasing time of flight with particle size (Supplementary Fig. 14) results in a complex variation in the particle preheat temperature (Fig. 1a–c) as determined through discrete particle method (DPM) simulations (Methods and Supplementary Note 6).

With increasing particle sizes from 10 μm until ~35 μm, the preheat temperature falls significantly with fine particles at very high temperatures (Fig. 1a). Exceeding a particle size of 35 μm, the preheat temperature increases with the particle size until it reaches a maximum of ~120 μm. Beyong this, the impact of particle size supersedes the time of flight and the temperature falls. Hence, depending upon the sizes of particles in a feedstock, the energy incident to the melt pool in form of preheated powders will vary and affect the melt pool formation. Also, with increasing laser power above 400 W, the particle preheat temperatures go beyond the evaporation temperature of SS316L. This would consequently result in vaporization of particles as they traverse through the laser and a decrease in the deposition of powders into the melt pool as evident from the reduction in deposition bead height and width for 500 W and 600 W single tracks (Supplementary Figs. 5 and 15).

For 300 W laser power (Fig. 1c), the fine PSD contains particles from region 1 where the temperatures reach the evaporation temperature for SS316L (~3000 K) hence the particles will be lost to vapourization whereas the coarse powder particles remain primarily in the superheated molten state. As the powder particles act as a secondary heat source to the melt, the additional power imposed by the fine and coarse powder feedstocks for a flow rate of 3.25 g min⁻¹ is determined to be 85.2 W and 92.0 W, respectively (Supplementary Note 4). A higher preheating of the molten coarse powder particles results in higher incident energy to the melt pool, which would affect its dimensions, their solidification parameters, and the resulting microstructural evolution.

### Application of particle size-dependent preheat temperature variations for site-specific microstructure control in PF-AM

We harnessed the powder-size dependent preheating effects by alternating between the fine and coarse powder feedstocks in-situ (Methods) to impart site-specific microstructure, as illustrated through a proof-of-concept (POC) part (Fig. 1d–f), printed using a constant set of process parameters (300 W laser power, 1000 mm min⁻¹ scanning speed, 3.25 g min⁻¹ powder flow rate, 0.2 mm layer height and 0.5 mm hatch spacing) by L-DED (Supplementary Fig. 2a) with the designed 'NTU' letters—visible after etching. The 'letter' zone in the POC part, denoted by L in Fig. 1e, was printed using the coarse powders while the 'matrix' zone, denoted by M in Fig. 1e, printed using the fine powders. The alternation between the fine to coarse powder feedstocks was done by executing a G-code that turned the hoppers containing the two powders on and off alternately while the deposition carried out unhindered. Notably, two distinct grain

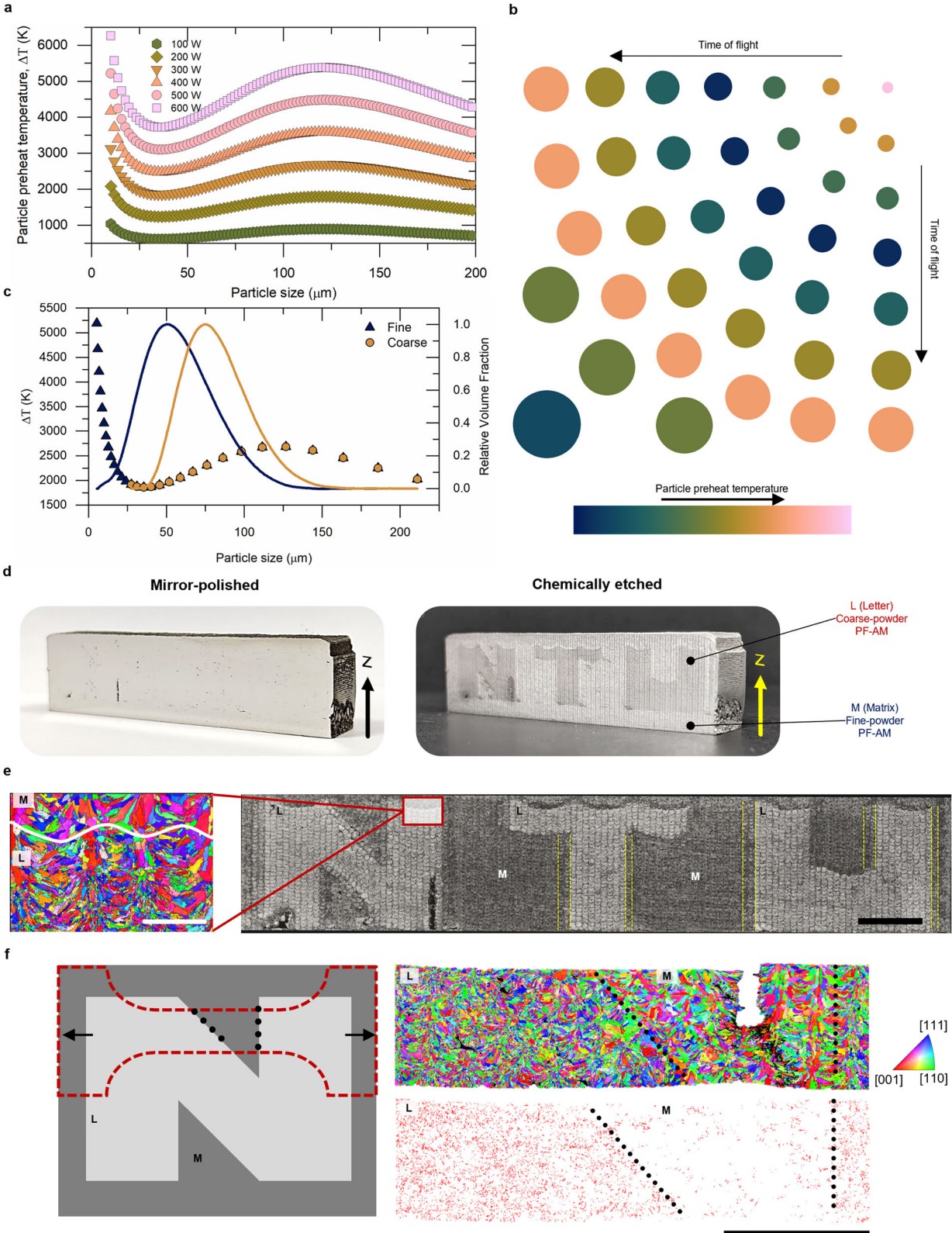

microstructure zones are observed along with transition regions enclosed by yellow dashed vertical lines. The resulting part showcases internal structure variation that is invisible when observed on a mirror-finished surface but reveals the three letters 'NTU' when etched. As seen in the inverse pole figure (IPF)-z map obtained from the EBSD, coarse elongated grains form in fine-powder 'matrix' and near-equiaxed FGs in coarse-powder 'letter' zone (Fig. 1e). The wider and

deeper melt pool topologies observed in the letter zone result in the growth of fine-equiaxed grains vs coarse-columnar grains observed in the matrix zone. The crystallographic orientation map associated with the etched micrograph highlights the differences in grain sizes in the two distinct regions that are demarcated by a white-colored wave-like boundary. This shows the efficacy of our MPE approach in facile site-specific microstructure control in 3D with the same material in

**Fig. 1 | PSD-driven grain morphology and size control in 3D-printed SS316L microstructure. a** Particle preheat temperature variation with particle size and laser power. **b** A schematic highlighting the time of flight and temperature variation with the particle size. Color of the particles corresponds to the temperature (Teal to Pink on the Batlow color palette[52] signify variation from low to high temperatures, respectively). **c** The particle size preheat temperature variation obtained for 300 W laser power and the PSDs measured for fine and coarse powders. **d** Sophisticated grain microstructure control achieved in this work with a proof-of-concept (POC) part via particle-size dependent MPE approach. **e** Melt pool and microstructural

information of the POC part. The associated scalebar is of length 5 mm. Magnified view of the horizontal interface of the matrix (M) and letter (L) zones of the letter 'N' shows the crystallographic orientation map (IPF-z) obtained using electron back-scattered diffraction (EBSD). The associated scale bar is 500 μm long. **f** Dogbone outline overlaid on the schematic of letter 'N' with the outward pointing arrows highlight the tensile coupon alignment in the sample. EBSD crystallographic orientation and twin boundary distribution maps obtained in the deformed region show the distinct response of the fine and coarse powder regions to deformation. The scale bar for large-scale EBSD maps of the tensile gauge section is 2 mm.

L-DED. The uniaxial tensile testing results from the N-section of the POC part are shown in Fig. 1f, to highlight the distinct deformation mechanisms in the L (letter) and M (matrix) regions (demarcated by black dotted lines). The IPF-z map of the gauge section showcases the fracture initiation at the M region with an associated map showing a significantly higher number of twin boundaries (in red) in the L region composed of fine-equiaxed microstructure.

## Feedstock PSD and variations in L-DED microstructure

Facile yet precise microstructure control in the L-DED process within the same layer is achieved in the POC part. We showcase that the PSD of the feedstock influences the melt pool architecture, and consequently, the grain microstructure, given the rise in particle temperatures as they pass through the region of laser irradiance and impinge into the melt pool.

Distinct grain sizes and shapes obtained in SS316L single tracks– the basic unit of an AM microstructure, printed using a set of 18 process parameters (Supplementary Table 1) by L-DED process for both fine- and coarse-powder feedstocks (Fig. 2, Supplementary Figs. 5–7). Charting the variations in the grain sizes and aspect ratios with the process parameter sets (Fig. 2a) reveals that overall, fine-equiaxed grains are obtained in the coarse-powder beads (Supplementary Note 2). Moreover, the grains in the equiaxed regions for the fine powder are still coarser than those obtained in the coarse-powder single tracks. We employed two melt parameters (Fig. 2b) namely: dilution (depth (d)/height (h)) and dimension (height (h)/width (w)) to chart the melt pool shape differences obtained for fine and coarse powders with laser power, scanning speed, and powder feed rate (Fig. 2c–e). We observe that the dilution of beads deposited using coarse granulated feedstock is larger for power >300 W (Fig. 2c) whereas an inverse behavior is observed from the dimension vs power curve. The gap in the dilution and dimension for the two feedstocks increases gradually with power before stagnating after 500 W. Moreover, the dilution of coarse-powder feedstock is larger than that of fine powders for the range of scanning speed of 500–3500 mm min⁻¹ (process parameter set 7–12 (Fig. 2d)). Likewise, the inverse behavior was observed in the dimension vs scanning speed curve, and the gap in dilution for the two feedstock increased gradually. The bead size of coarse powder deposition is larger than that of fine powder ones (Fig. 2e).

A larger dilution implies deeper melt pools or smaller bead heights and vice-versa. While a larger dimension indicates taller or narrower beads and vice-versa. This fundamental study with L-DED also highlights that the melt dilution measured for the coarse-powder single tracks increases almost linearly with power (Fig. 2c) and laser scanning speed (Fig. 2d). It is suggested that even at higher power and scanning speed, coarse-powder melts will penetrate deeper with more equiaxed and finer grains in comparison to their fine-powder counterparts.

## PSD-driven microstructural variations applied to L-DED stainless steel samples

We confirmed the opposing microstructural response of particle sizes in L-DED single tracks. To verify the powder-size driven MPE approach, we fabricated L-DED block samples using fine, a mixture of fine and coarse (FC) (mixing ratio 1:1), and coarse powders (Methods and

Supplementary Fig. 2b). We found that both the melt pool depth and width, measured in the transverse plane, increase with increasing average particle size (Fig. 3a–c). The melt pool depth measurements for fine, FC, and coarse powder samples are 380, 407, and 458 μm, respectively. The same for melt pool width is obtained as 657, 675, and 712 μm, respectively. Substantial epitaxial grain growth is noted in the fine powder sample (Fig. 3a) with coarsening of the well-oriented grains. As predicted, the FC microstructural evolution seen in Fig. 3b resembles closely to that of the coarse powder one shown in Fig. 3c with slightly elongated grains directed towards the centre of the melt pool. We also observe that CET occurred in the microstructure of 3D-printed parts using coarse and FC powders, which is consistent with our assumptions on the grain morphological evolution as confirmed by the grain statistics quantified from the EBSD maps (Fig. 3d). We discover that the FC powder feedstock induces a microstructural evolution where the grain sizes and their aspect ratios lie midway to those obtained using fine and coarse powders, roughly closer to the latter, since it is primarily inclusive of the particles with sizes in the range of fine-powder PSD, which is evident from the powder flow rates provided in the Supplementary Fig. 2b. Average grain widths of 30.2, 15.6, and 13.1 μm are obtained for fine, FC, and coarse powder samples, respectively, with average aspect ratios of 2.5, 2.4, and 2.2, respectively.

## PSD-driven microstructure control in PBF–an innovation over conventional approaches for CET in metal AM

We envision that PSD-driven microstructural variations would be effective in producing differences in microstructures in cases where the particle size could affect the overall cooling process–such as in the case of PBF AM processes. Given the limitations of commercial L-PBF systems to process particle sizes in the range of 15−63 μm, we chose the E-PBF process to establish the facility of our approach for producing fine-equiaxed microstructure in a technique that has been widely established to produce near-monocrystalline microstructures in SS316L[32] and single-crystal microstructures in Ni-based superalloys[33,34]. Though the application of our approach to L-DED for site-specific microstructure control was intuitive–supported by our analytical calculations and numerical simulations, same is not as straightforward when applied to the E-PBF process due to the complexities associated with the effect of PSD on the powder-bed thermophysical properties. To advance the state of the art, we developed a machine learning (ML) framework assisted by particulate microstructure modeling to investigate the impact of PSD on the powder-bed thermophysical properties which will be detailed in the following section.

## Machine learning framework

We identified a research gap in relating the impact of PSD on microstructure and mechanical properties for PBF systems. It is likely due to the lack of a framework that could provide predictability of powder-bed thermophysical properties, specifically powder-bed density (PBD) and thermal conductivity, from the feedstock PSD. For this purpose, we employed machine learning (ML) coupled with numerical simulations of heat transfer in particulate microstructures, representative of sintered powder bed. Figure 4a displays the ML framework built for the determination of feedstock PSD that would possibly result in a wide

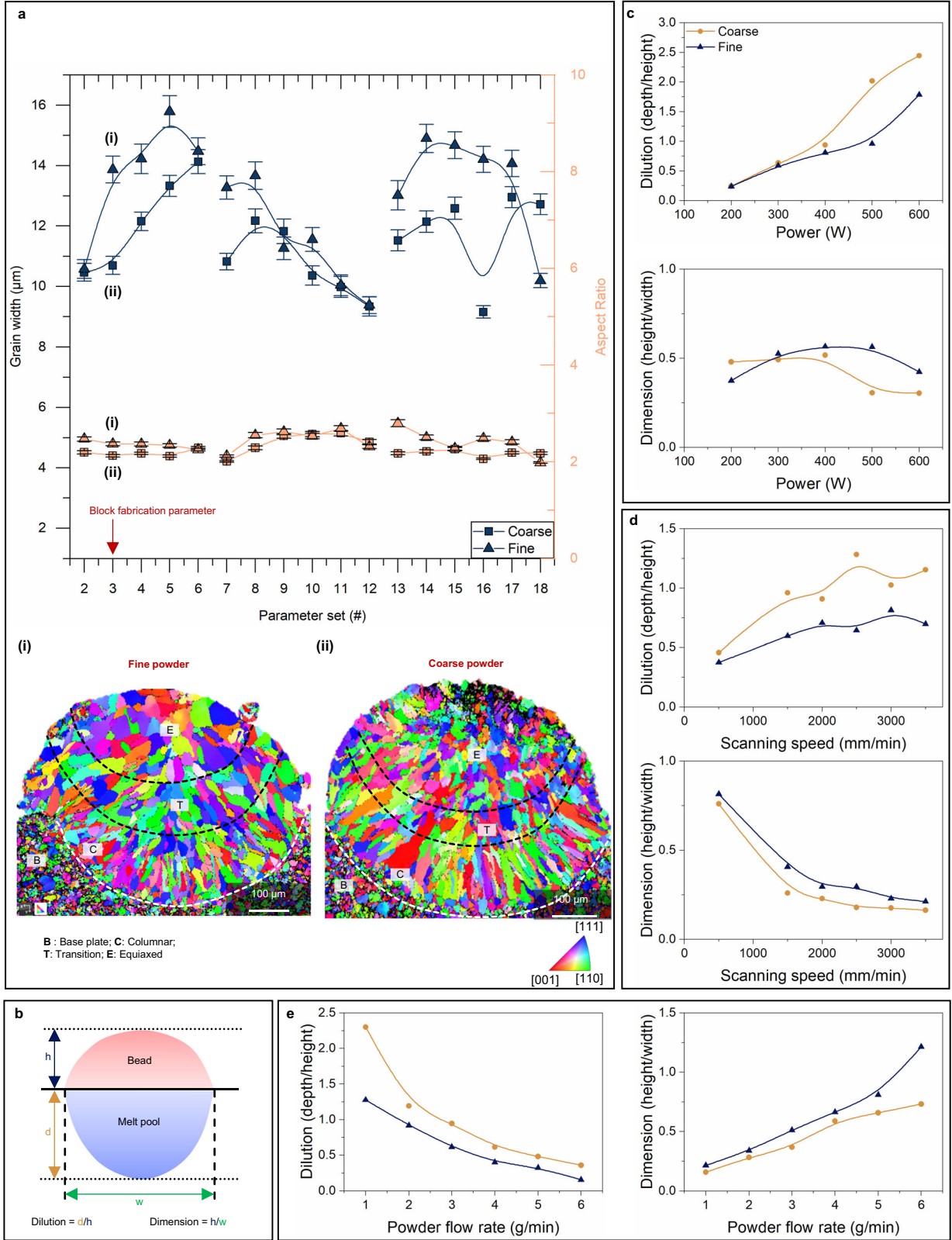

**Fig. 2 | Melt pool architecture and grain morphology in DED printed single tracks of SS316 L. a** Variation in the grain widths and aspect ratios obtained from single-track depositions for process parameter set 2–18. Error bars represent ±standard error of the mean. (i) and (ii) are the IPF-z maps obtained from the transverse sections of single tracks deposited using parameter set # 3 (block fabrication parameter) using fine and coarse powders, respectively. White dashed lines in the maps demarcate the melt pool boundary. Columnar, equiaxed, and transition regions per visual inspections of the single-track beads have been distinguised using black colored dashed lines. **b** Schematic of a deposited bead with relevant dimensions and nomenclature. h, d, and w – are notations for bead height, melt pool depth, and melt pool width, respectively. **c** Variation of dilution and dimension vs laser power. **d** Variation of dilution and dimension vs laser scanning speed. **e** Variation of dilution and dimension vs powder flow rate.

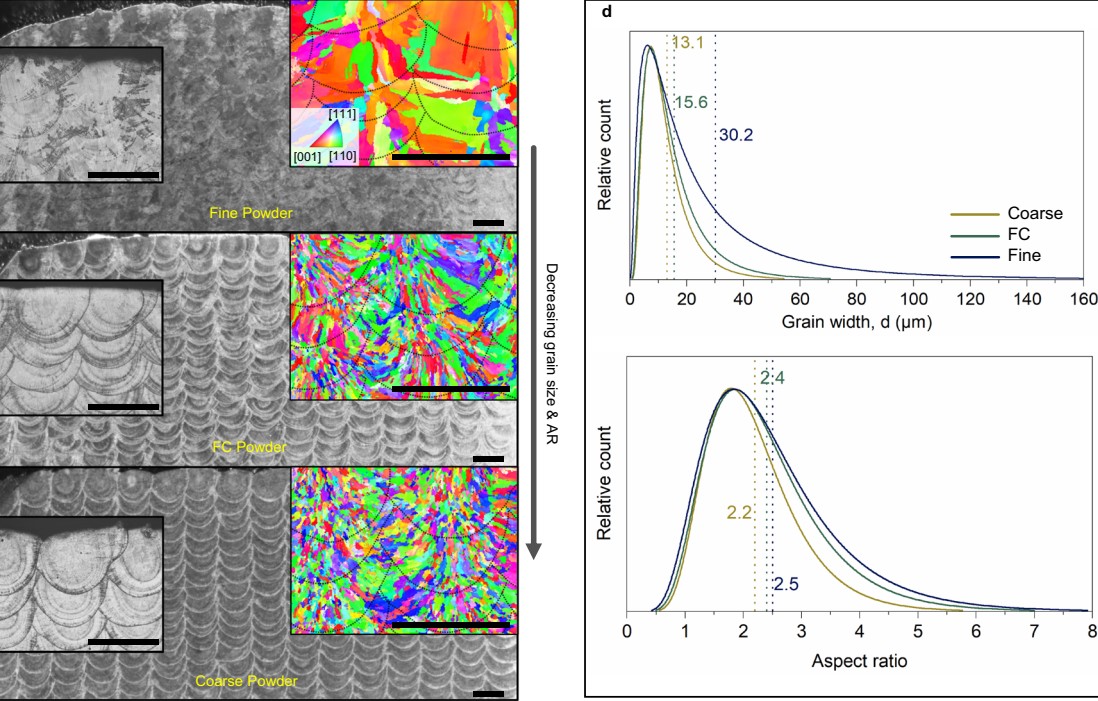

**Fig. 3 | PSD-driven microstructure control in DED process.** Melt pool/Microstructural observations in (**a**) Fine, (**b**) FC, and (**c**) Coarse powder as-built samples. All of the scales correspond to 500 μm. The EBSD maps taken along the build direction in the inset showcase the variation in grain morphology and orientation with increasing PSD. **d** Quantified measurements of the grain width and aspect ratio obtained for the fine, FC, and coarse powder samples.

range of microstructures (Supplementary Note 5). Twenty-six PSDs were employed to perform particulate microstructure modeling in Mote3D[35]. The simulated particulate microstructures were added to the ABAQUS FEA environment for performing 1-D heat conduction simulations. Thermal conductivity and powder bed density thus obtained were fed to a ML model, and the result is shown in the provided response curve (Methods). The ML response curve obtained for the PBD and the thermal conductivity (Fig. 4b) showcases both the limitations of the literature in assessing the impact of PSD on the microstructure and the mechanical properties of 3D-printed samples as well as the advancement that our approach can bring to the state-of-the-art PBF systems. The contour plot, with an assumption of normal size distribution in the particle size statistics, indicates the variation in the relative powder bed density (PBD) from 25 to 75% concerning the mean particle size and standard deviation. The overlaid solid lines show the variation of thermal conductivity in the intervals of 1, 2, 3, 4, and 5 W m$^{-1}$K$^{-1}$. R1, 2, and 3 are three regions of interest (ROIs) and the stars indicate the fine (empty) and coarse (filled) powder statistics employed by us for part fabrication. Our ML model confirms that there is no linear relationship between PBD and the thermal conductivity (k) of the powder bed as seen from the contours. To some extent, we can confidently state that the ML curve can predict the microstructure and mechanical response variations obtained with PSDs reported in the literature[24,28]. Among the three ROIs in Fig. 4, R1 is the region in which the PBD is less than 37.5, in R2 it is greater than 66.7 and in R3 it is again less than 37.5. Both R1 and R3 are regions where PBD is less than 37.5 but in R1 there are no drastic changes in the PBD with the variations in mean and deviation in the particle size. In fact, in R1, the PBD varies from ~35 to 37.5% for a mean particle size range of ~20–40 μm. The same is not true for R3 where for a mean particle range of ~100–120 μm or a deviation range of 15–50 μm the PBD varies drastically in the range of ~25–37.5%. One can identify two regions on the ML response curve where the k value is between 0 and 1 W m$^{-1}$K$^{-1}$. The first region, which also covers part of R1, extends from a mean particle size of 20 μm to that of 94 μm for some varying deviation values. Interestingly, the k

value from this region increases along the opposite directions of mean particle size shown by the white arrows. The fine and coarse PSDs investigated in the literature[24,28] can be located on the ML curve in circle symbols[28] and inverted triangles[24] with the help of their respective particle size statistics (D10, D50, D90). We can dictate that the coarser PSD (red inverted triangle and circle) from the literature are located such that they have lower powder-bed thermal conductivity than their fine PSD counterparts. This is the likely reason for the coarser PSD investigated resulting in similar or higher PBD for a comparatively lower mechanical property due to the solidification driven by an inferior thermal conductivity. These points also show the extent to which PSD's effect on the microstructure and mechanical response has been investigated in the literature and the significantly large domain that remains to be explored. The PSDs of the fine- and coarse-powder feedstocks employed by us for the L-DED fabrications (denoted by white and blue stars in the response curve, respectively) based on their descriptive statistics fall on low and high thermophysical regions, respectively, of the curve and hence are ideal to validate the ML model.

### Effect of feedstock PSD on powder-bed thermophysical properties in E-PBF SS316L

A packing density of 81% is obtained for the coarse powder bed, which is higher than that of 77% for the fine powder bed quantified from the μ-CT measurements (Fig. 4 c and d). The thermal diffusivity values at 850 °C were measured as 0.74 and 0.87 mm$^2$ s$^{-1}$ for the fine- and coarse-powder beds, respectively, using a laser flash apparatus (LFA) (Methods and Supplementary Note 1). The same was determined as ~6 mm$^2$ s$^{-1}$ at 850 °C for as-built bulk samples. A particle-size independent specific heat capacity value ~ 565 J kg$^{-1}$K$^{-1}$ was also obtained for both fine and coarse powder-bed samples at 450 °C. The temperature-dependent data obtained from these measurements and the fitted curves can be seen in Supplementary Fig. 3. From these, the estimated thermal conductivity values are 1.2 and 2.0 W m$^{-1}$K$^{-1}$ at 25 °C, and 1.8 and 3.0 W m$^{-1}$K$^{-1}$ at 850 °C for fine- and coarse-powder beds,

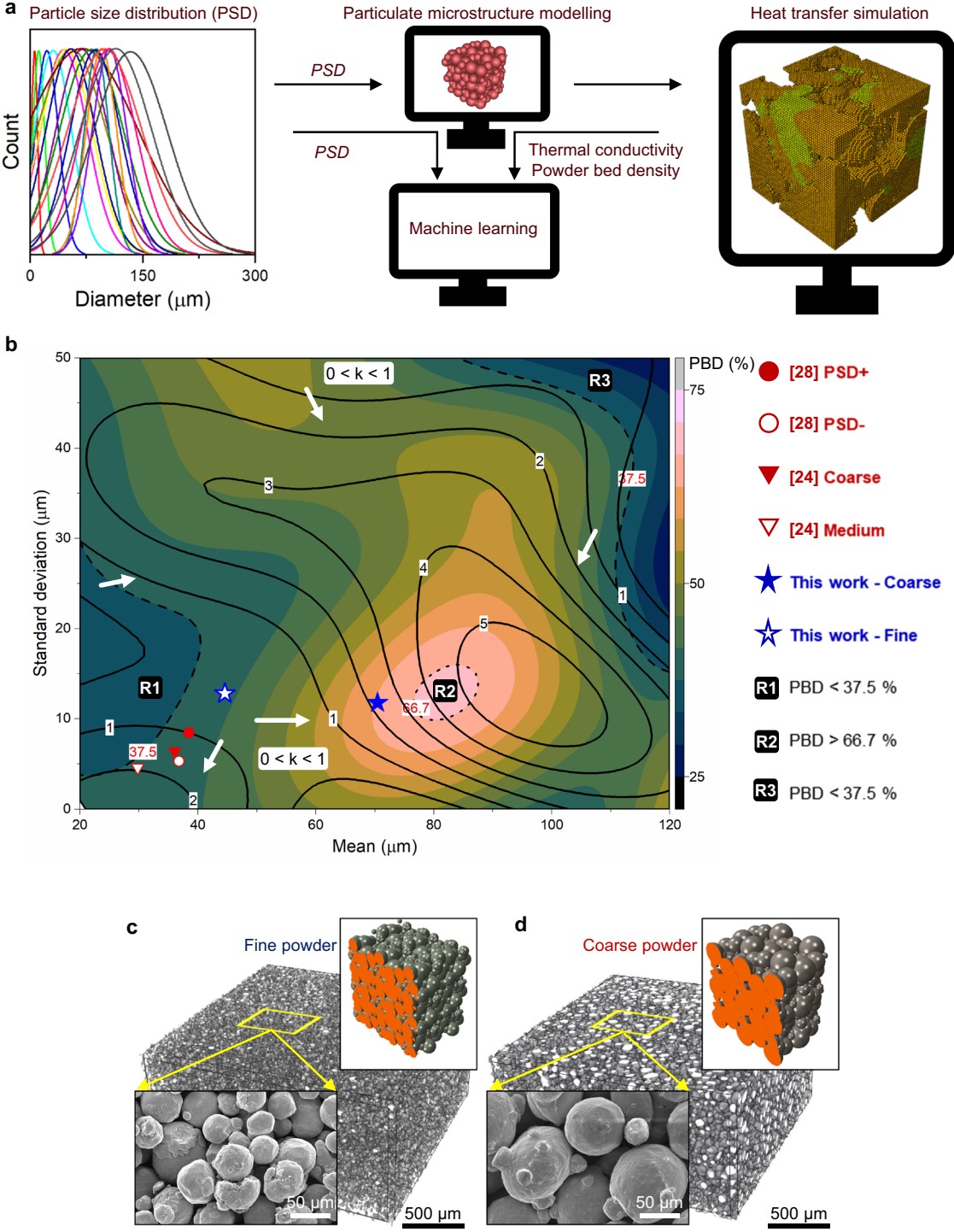

**Fig. 4 | ML framework employed to predict the impact of PSD on the thermo-physical properties of a sintered powder bed. a** ML framework presenting the flow of data from simulated particulate microstructure to the FEA simulations and the ML algorithm. **b** Response curve for thermal conductivity (*k*) and powder bed density (PBD) obtained from the ML framework. White arrows indicate the direction of increasing *k* on the response curve. Representative 3D rendering of (**c**) fine and (**d**) coarse sintered powder beds analysed using μ-CT scanning, respectively, along with their respective simulated particulate microstructure. The inset images are FESEM micrographs of fine and coarse sintered powder beds taken at the same magnification.

respectively. Though there is some disparity in the PBD measurement vs prediction from the ML response curve shown in Fig. 4b, it can be justified through the consistent employ of D50 as a representative in the ML curve which is developed for mean and standard deviation. The ML curve provides an accurate estimation of the thermal conductivity of the coarse-powder bed and underestimates the same for the fine-powder bed.

## Engineering CET in E-PBF 316 L stainless steel samples

From our earlier research on E-PBF SS316 L[8,32], we found that fine-powder feedstock under optimum process parameters for printing fully dense samples resulted in a coarse-columnar microstructure which showed obvious mechanical anisotropy. Through our powder-size driven MPE approach (Methods), noticeable CET is achieved from the fine- to coarse-powder samples, which is evident from their

respective EBSD micrographs (Fig. 5a, b). The near-monocrystalline coarse-grained microstructure is obtained in the fine-powder E-PBF sample along with wide and shallow melt pool contours (w/d: 5.6) in the transverse plane. In stark contrast to this, a diffuse microstructure consisting of substantially fine grain sizes is obtained in the coarse-powder samples with semicircular melt pool contours (w/d: 2) in the transverse plane.

The coarse-columnar microstructure in the fine-powder samples has an aspect ratio of $10.9 \pm 1.1$ and a grain width of $27.6 \pm 2.8\,\mu m$ (Fig. 5c). On the contrary, the FG microstructure is observed in the coarse-powder samples with $9.2 \pm 0.5\,\mu m$ wide which can be attributed to the heterogeneous nucleation that occurred during the melt pool solidification. This near-equiaxed, FG microstructure, with an AR value of $2.7 \pm 0.1$ is the reason behind the superior directional isotropy observed in the mechanical properties of the 3D-printed samples (Fig. 6b). From the EBSD maps, the grain sizes in the fine- and coarse-powder feedstocks were determined as $2.3 \pm 0.2\,\mu m$ and $6.2 \pm 0.4\,\mu m$, respectively (Supplementary Fig. 8). Even though the coarse-powder

feedstock had a larger grain size in comparison to the fine powders, an inverse trend in the grain size of 3D-printed samples could be obtained by our approach. The formation of these FG and near-monocrystalline microstructure in coarse and fine powder E-PBF samples can be explained by the schematic shown in Fig. 5d. It was determined through LFA testing and FEM simulations on a granulated feedstock that the fine PSD exhibits a lower powder-bed thermal conductivity and a lower PBD in comparison to the coarse PSD during the E-PBF process hence the heat transfer in the latter through the deposited material and the sintered powder bed is comparable and higher than that in case of the fine PSD. A much higher energy density is required by the E-PBF system in place to maintain the desired build temperature that results in a convection-type melt pool geometry causing the CET.

### Solidification maps for the powder-size driven MPE in L-DED and E-PBF processes

We have shown that as we had hypothesized, the powder sizes (or PSD) have shown to influence microstructural evolution in both the L-DED

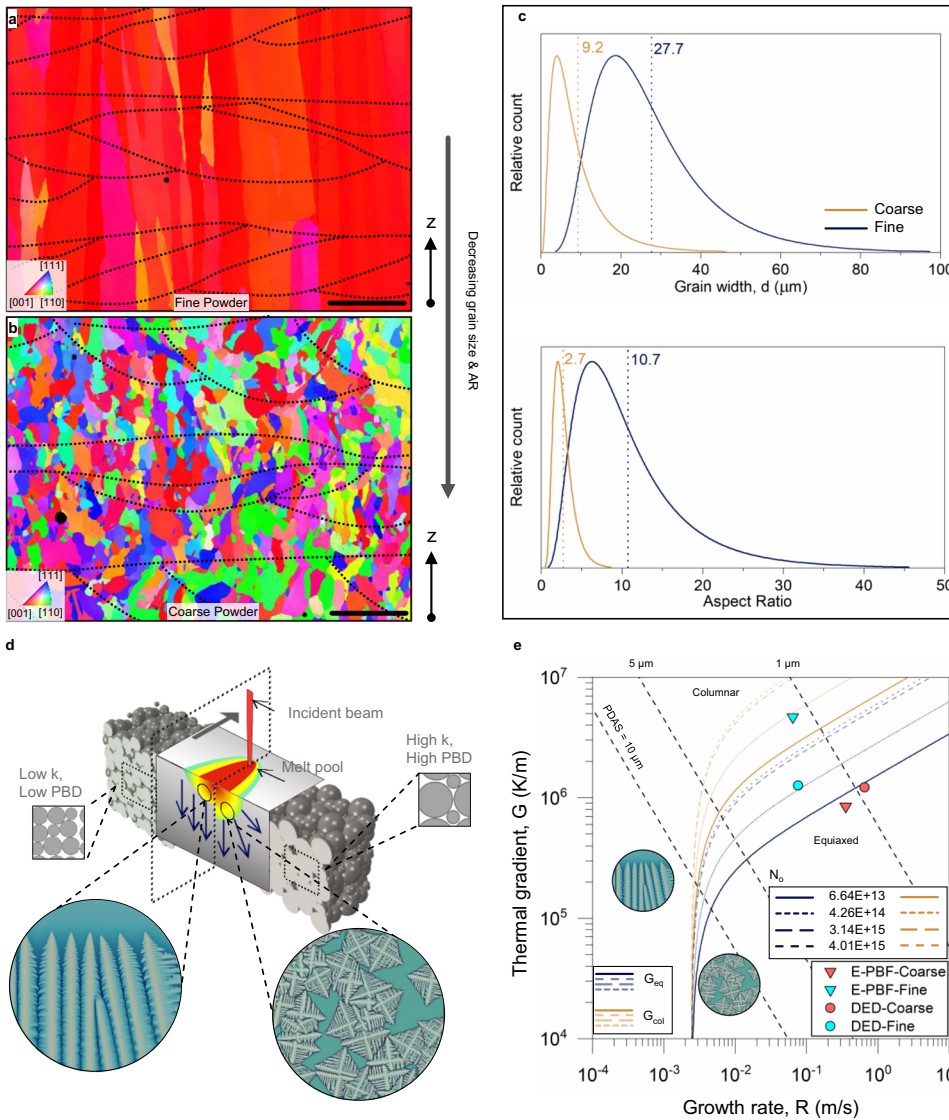

**Fig. 5 | Powder-size driven MPE applied to E-PBF process for a facile and large-scale grain control.** IPF-z maps obtained from EBSD of E-PBF fabricated (**a**) fine- and (**b**) coarse-powder samples. The scale bars in both (**a**) and (**b**) are 100 μm. **c** Quantitative analysis of grain size and shape obtained in the fine- and coarse-powder samples. **d** General schematic on the impact of coarse vs fine PSD on the melt pool shape and its microstructure. **e** Solidification maps with distinct

columnar and equiaxed regions (demarcated by orange and blue boundaries, respectively). Four different values of nucleation density, $N_o$, of $6.64 \times 10^{13}$, $4.26 \times 10^{14}$, $3.14 \times 10^{15}$, and $4.01 \times 10^{15}$ are determined through experimental observations of equiaxed grain volume fractions obtained in the E-PBF-Fine, E-PBF-Coarse, L-DED-Fine, and L-DED-Coarse EBSD maps. PDAS stands for primary dendrite arm spacing.

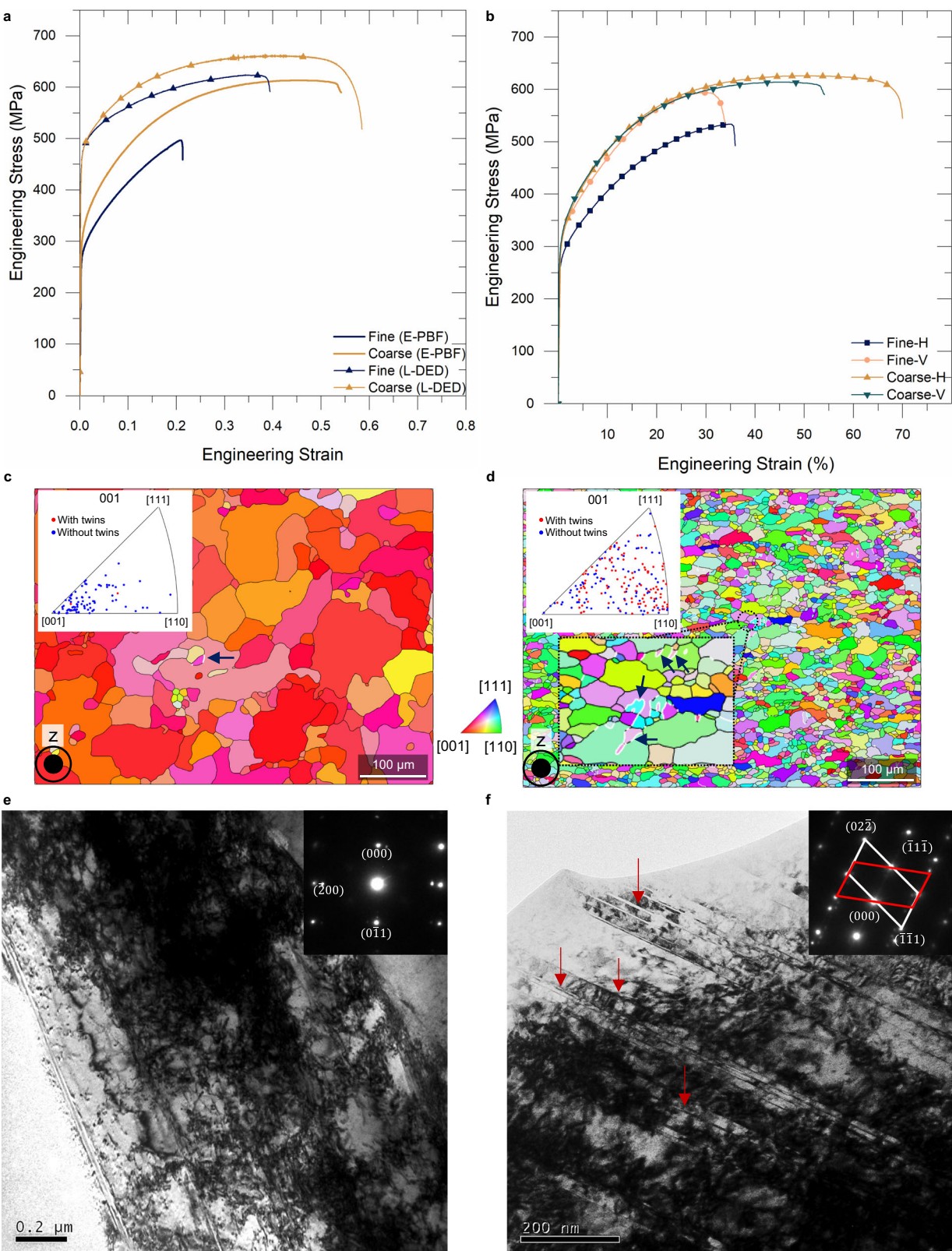

**Fig. 6 | Mechanical response of E-PBF and L-DED fabricated fine- and coarse-powder samples. a** Tensile response curves obtained for L-DED and E-PBF samples tested along the build direction. **b** Engineering stress vs strain (%) obtained for E-PBF samples highlights the improvement in mechanical isotropy attributed to the FG equiaxed microstructure obtained in the coarse-powder samples. H indicates horizontal direction and V indicates vertical direction (or build direction). **c**, **d** are IPF-z maps obtained for fine- and coarse-powder samples, respectively, taken 3 mm

away from the fracture surface. The twin grain boundaries observed in the EBSD maps are colored in white and pointed by dark blue arrowheads. **e** TEM bright field image of fine powder sample showing stacking faults. **f** TEM bright field image of coarse-powder samples showing a large number of deformation nanotwins, indicated using red arrows. Inset figures in (**e**, **f**) are their corresponding selected area electron diffraction (SAED) patterns obtained using TEM. The tensile testing direction with respect to (**c**–**f**) was along the plane of the figure.

and E-PBF processes, though more substantial results are obtained in the E-PBF process due to the vital role of the powder bed in the melt pool solidification coupled with our inventive E-PBF manufacturing strategy for the two powders (Methods). Furthermore, the examination of the solidification conditions occurring in the fine- and coarse-powder samples printed using L-DED and E-PBF in perspective of the CET curve[10,36,37] is the key to revealing the underlying mechanisms of our approach. Figure 5e presents the CET curve obtained for SS316L (Supplementary Note 8), where the points conforming to the solidification parameters obtained for the coarse-powder samples for both L-DED and E-PBF lie in the equiaxed region while the same in the columnar region for the E-PBF fine-powder samples. The solidification parameters for the L-DED fine-powder samples are still in the equiaxed region which can also be confirmed from the microstructural investigations presented in Fig. 3a and the grain statistics provided in Fig. 5c. We employed numerical simulation techniques for the determination of thermal gradient ($G$) values in the melt pools of L-DED and E-PBF and analytical correlation[38] for the estimation of *cooling rate* from primary dendrite arm spacing (PDAS). The values of $G$ and *cooling rate* together can determine the solidification rate ($R$) all of which are then employed to pinpoint the CET mechanisms in the L-DED and E-PBF processes for the fine- and coarse-powder feedstocks (Supplementary Note 8). We found out that under the current E-PBF processing conditions, CET in coarse-powder samples occurred due to the lower $G$ and higher $R$ in melt pools subjected to similar cooling rates. In L-DED coarse-powder samples, however, lower $G$ was also accompanied by a higher *cooling rate* resulting in a substantial reduction in the average grain size and PDAS.

## Mechanical response of varying microstructures tuned by powder-size driven MPE approach

The representative engineering stress vs strain curves obtained from the uniaxial tensile testing for the E-PBF and L-DED fabricated SS316 L samples (Fig. 6a) underline the distinct mechanical response of the coarse and fine microstructures obtained using the fine- and coarse-powder feedstocks, respectively. Importantly, they highlight the need for achieving as fine a microstructure as possible if the desire is to obtain unprecedented mechanical response, especially in the case of 3D-printed stainless steel. The coarse granulated feedstock resulted in better properties in E-PBF and L-DED alike. The fine-powder samples obtained from the E-PBF process have an average grain width of ~28 μm and an AR value of ~11. This resulted in the most inferior properties, which is primarily attributed to the coarse-columnar microstructure and partly due to the formation of detrimental σ precipitates at the columnar grain boundaries reported in an earlier work[8]. The E-PBF printed samples exhibit significantly lower yield strength (YS) values compared to L-DED due to the in-situ annealing associated with the former. Another key takeaway from the uniaxial tensile testing of the E-PBF samples is the fascinating level of mechanical isotropy obtained in the coarse powder samples, presented in Fig. 6b, where a yield strength value of ~310 MPa was obtained both along the horizontal (H) and vertical (V) directions with a deviation of ~0.6% in the ultimate tensile strength (UTS). To our best knowledge, this level of isotropy in mechanical strength is substantially superior to that reported in the literature[6–8,39–49], and it can be attributed to the near-equiaxed, fine-grained microstructure in the 3D-printed coarse-powder samples. The L-DED fabricated coarse-powder samples, on the other hand, provide a superior mechanical response overall in terms of YS, UTS, uniform elongation (UE), and elongation to fracture ($\varepsilon_f$). We believe that a reduction of ~57% in the grain width of coarse-powder samples from the fine-powder ones is the driving force behind a high UTS of 664.5 MPa (662.4 ± 1.9 MPa) for UE of 38.4% (38 ± 0.5%), and $\varepsilon_f$ of 58.5% (56 ± 1%). UTS, UE, and $\varepsilon_f$ values of 624.4 (629.4 ± 2.7 MPa), 34.4 % (34 ± 1%), and 39.4% (42 ± 3%), respectively, are obtained in fine-powder samples. From an earlier report[50], we can confirm that our

coarse-powder samples have shown one of the best UTS and $\varepsilon_f$ combination among the 3D-printed SS316L samples so far.

Figure 6c, d show the comparative EBSD analyses of deformed fine- and coarse-powder samples, respectively. The ROIs were chosen 3 mm from the fracture surface. The tensile testing direction resides in the plane of the maps and is perpendicular to the build direction, z. The colors associated with each grain correspond to the IPF scale attached to the figures. The <001> directions of most of the grains in fine-powder samples are oriented along the build direction. While the crystallographic orientations of the grains in the coarse-powder samples are random—like the as-built microstructure. Many deformation twins (white boundaries showed with blue arrows) are observed in coarse-powder samples as seen in the magnified overlay, while, within the ROI, twin boundary is rarely seen in the fine powder sample. The overlayed IPF shows the grains with and without twin boundaries. It is seen here that the fractured coarse-powder specimens have twin boundaries associated with nearly half of the total grain population. Moreover, twins are observed in grains which even have orientations close to <001 > . However, this is not true in the case of fine-powder samples. Similar observations were reaffirmed by the TEM investigations of deformed fine- and coarse-powder samples presented in the Fig. 6e, f, respectively. Bright-field (BF) TEM image along the <011> zone axis (Fig. 6f) shows substantial nano-scale twins (nano twins) which are marked by red arrows in the coarse powder sample. The SAED pattern shows typical streak lines and can be resolved into two distinct patterns mirrored along the <111> in-plane direction which confirms the presence of deformation twins. The BF image of the deformed fine powder sample (Fig. 6e) also contains a few planar defects. However, the presence of a large number of dislocations in the region and the lack of twin patterns in the associated SAED pattern confirm these to be stacking faults (SF). Hence, it can be known that the plastic deformation in the coarse-powder samples under uniaxial tension was largely coordinated through nanotwinning, whereas dislocation slip prevailed in the fine-powder samples.

## Discussion

Our facile and large-scale microstructure control is accompanied by the CET in melt pool solidification. We showcase CET in both powder-flow L-DED and powder-bed E-PBF with a drastic transformation from columnar to equiaxed grain microstructure with fine grain sizes roughly 3 times finer (9.2 ± 0.5 μm) than the former in E-PBF deposited SS316 L. As an added advantage, the equiaxed microstructure resulted in an isotropic mechanical response with the UTS and elongation ~17% and ~62 % higher, respectively, than the coarse-columnar microstructure by twinning induced plasticity (TWIP) (Fig. 6). This study has presented facile, low-cost yet sustainable ways for controlling the grain microstructure in 3D-printed metal parts. The observations regarding the impact of feedstock PSD on the melt pool geometries in two of the mainstream PF-AM techniques can prove fruitful in furthering the material scope of metal AM. In the L-DED process, particle size governs its flow through the laser irradiance zone, hence its preheating acts as a secondary heating source to the melt pool. Whereas in the E-PBF process, the variation of thermophysical properties of the powder bed drives the microstructural transformations. With the recent advent of high-power L-PBF processing that reports employ of coarse powder sizes ~90 μm[29], our powder-size driven MPE approach has the potential to adapt to the widely popular L-PBF process as well. The ML framework developed by us highlights the complex relationship between PSD and the powder-bed thermophysical properties. Moreover, it is suggested that substantial research is needed to explore larger particle sizes than the current norm.

In summary, we not only demonstrated the effect of the PSD of the granular feedstock on the AM microstructure but also explored it to produce site-specific microstructure control. The final set of samples from both fine and coarse powders preserved the original

chemical compositions of SS316L. We realized facile control in grain microstructure, i.e., a wide spectrum of grain morphology and size that had been scarcely achieved. The near-equiaxed FG microstructure of our 3D-printed samples is one of a kind, and it opens the applicability of a conventional alloy such as SS316L to extraordinary possibilities such as high strength and ductility, mechanical isotropy and homogeneity, and superplasticity. On the contrary, the near-monocrystalline microstructure obtained from fine powders provides guidance to print Ni-based superalloy single crystals for desirable high-temperature creep resistance.

## Methods

### Materials

Two different batches of pre-alloyed SS316L powders, provided by TLS Technik GmBH, were utilized in this study. The cast billets of SS316L were melted in vacuum and then atomized using argon gas for both fine and coarse powders. The nominal elemental compositions of the pre-alloyed powder were Fe-11.8Ni-17.1Cr-2.3Mo-0.4Mn-0.02C-0.8Si (wt.%). The actual elemental compositions measured using the ICP-OES (Agilent 720 series analyzer) and combustion methods (Eltra CS800 and ONH2000 analyzers) are provided in Supplementary Table 5. These powder batches exhibited different PSDs as well as morphologies, as shown in Supplementary Fig. 1, and were classified as fine (Supplementary Fig. 1a) and coarse (Supplementary Fig. 1b) powders depending upon their particulate size arrangement. Since PSD is crucial to the SS316L fabrication in this research, a laser particle size analyzer—Malvern Panalytical's Mastersizer 3000 (Supplementary Fig. 1c), was utilized to measure the same. For this purpose, the powder sample was first loaded into the Aero S adapter's hopper. The powder then vibrated slowly down through the channel feeder to the laser particle-size analyzer. The feed rate for the sample feeder was set to 10%, and the air pressure used for the analysis was set to 4 bar. A laser beam was passed through the dispersed powder sample, and, as a result, the incident light was scattered at small angles by large-sized powder particles and high angles by small-sized ones (Supplementary Fig. 1d). The angular variation in the intensity of light was analysed to calculate the size of the particles according to the Mie scattering[51] and reported as the diameter of a volume-equivalent sphere. The PSD obtained from the Mastersizer is provided in Supplementary Fig. 1e, f, for fine- and coarse-powder feedstocks, respectively. The D10, D50, and D90 values were measured for the fine-powder batch as 23 μm, 45 μm, and 75 μm, and the same as 50 μm, 70 μm, and 100 μm for the coarse-powder batch. Throughout this article, we have presented these valuable size statistics as (D10, D50, D90). For the printing with the DED process, another batch of feedstock was prepared by blending of fine and coarse powder in a 1:1 weight ratio. The powders were mixed in a tumbler mixer, Inversina 2 L, at a tumbling speed of 30 rpm for 2 h. The resulting mixture of fine- and coarse-powder particles was termed FC powder throughout the main and supplementary text.

Stainless steel 304 baseplates (150 mm × 150 mm × 10 mm) were used for the fabrication of block SS316L samples using both L-DED and E-PBF processes. SS316L baseplate was used for the deposition of single tracks using the L-DED process.

**DED process depositions.** Beam Magic 500 DED machine was employed in this work to print the POC part (Fig. 1), single tracks (Fig. 2), and block samples using fine, coarse, and FC feedstocks (Fig. 3). The POC part and the block samples printed using the same process parameters are detailed here. An Nd:YAG fibre laser module with a maximum power of 1 kW was employed for fabrications with laser spot size and standoff distance of 0.75 mm and 3.5 mm, respectively. For the fabrication of the POC part, the two powder hoppers of the DED system were filled with fine and coarse powder feedstocks

separately. A toolpath file (provided upon request) was generated that activated the fine- or coarse-powder hoppers by selectively controlling the feedstock type for every deposited bead in the layer of the part as per the designed part geometry (Supplementary Fig. 2a). As a compulsory preliminary step, the hopper turntable speeds were adjusted to achieve the optimum 3.25 g min$^{-1}$ of powder flow rate ($P_f$) for each of the fine-, coarse-, and FC-powder feedstocks. This was achieved by plotting the powder flow rate (amount of powder in gms being collected in a container from the nozzle during 1 min of turntable rotation) vs turntable speed specified in percentages of the maximum turntable rotation speed (Supplementary Fig. 2b). For fine powder, it was finalized as 14.8% and 16.5% for coarse powder through extrapolation. The slower turntable speed for fine powder was due to the increased capacity of carrier gas to circulate the lighter fine powder to the laser spot. Argon gas was chosen as the carrier gas with a fixed flow rate of 3 litres min$^{-1}$. It is also employed as shielding and secondary gas flow with 3 and 6 litres min$^{-1}$ flow rates, respectively. Layer height (Δz) and hatch spacing (Δxy) of 0.2 mm and 0.5 mm, respectively, were chosen. Laser wattage (P) and scanning speed ($V_s$) of 300 W and 1000 mm min$^{-1}$ were chosen—constant for the three powders. Non-raster scanning was employed with a laser moving in and out of the plane of the figure. The dimensions of the 'POC' block were 20 mm long (along the laser deposition), 60 mm wide, and 15 mm tall. A separate block—20 mm wide, 20 mm long, and 15 mm high, having the letter 'N' aligned with the build direction was printed using the aforementioned process parameters for the extraction of tensile samples as shown in Fig. 1. For quantifying the microstructural differences between L-DED deposited fine-, coarse-, and FC-powder samples (shown in Fig. 3a–c) cuboidal samples 15 mm long, 10 mm wide, and 10 mm high were printed with the same process parameters as well. Samples 20 mm long, 20 mm wide, and 20 mm high were printed using fine and coarse powders—for fabrication of coupons for tensile testing.

For the deposition of fine-, coarse-, and FC-powder single tracks, power, scanning speed, and powder flow rate are varied for six parameters each resulting in a combination of 18 process parameter sets for which the single tracks were deposited, sectioned in the transverse plane, and measured for the melt pool characterization parameters of dilution (depth/height) and dimension (height/width). A judicious experiment was designed based on our experience of printing block samples with a wattage of 300 W, scanning speed of 1000 mm min$^{-1}$, and powder flow rates of 3.25, 6.5, and 9.75 g min$^{-1}$. Hence, a wattage of 100−600 W, scanning speed of 500−3500 mm min$^{-1}$, and powder flow rate of 1−6 g min$^{-1}$ were chosen to demonstrate the PSD-driven melt pool and microstructural evolutions. The complete set of process parameters employed for the fabrication of single-track specimens has been provided in Supplementary Table 1.

**E-PBF process depositions.** An ARCAM (now GE Additive) A2XX EBM® system was employed for the layer-wise fabrication of fine- and coarse-powder SS316L samples. An optimum set of process parameters determined through an earlier research[8] were provided to the EBM system for the printing process. A snake-type hatching pattern—a default hatching pattern in ARCAM's A2XX EBM® system, was employed for the fabrication of block samples that were used for microstructural and mechanical property characterizations. A speed function (SF)/current combination of 150/16 mA was provided as the user input which corresponds to the beam scanning speed of ~5400 mm s$^{-1}$ and 960 W beam power. The preheat temperature was fixed to 850 °C with a hatch spacing of 100 μm and a layer height of 50 μm. The printing process was carried out in the automated mode of the ARCAM EBM system with the assistance of its proprietary software – 1D analyze. The technique for obtaining FG equiaxed microstructure by coarse powders in E-PBF by harnessing this automation has been detailed in Supplementary Note 1.

**Metallography.** The 3D printed samples were sectioned using Troop's TP-50 electric discharge machining (EDM) wirecut system. The specimens obtained from the EDM were manually ground and subsequently polished to a mirror finish using Struer's LaboForce-50. The grinding process was carried out using SiC papers of grit sizes 200, 600, 800, 1000, and 2000. Polishing of the specimen surfaces to a mirror finish (akin to that shown in Fig. 1) was carried out using DiaPro Dac (3 μm suspension) and MD-Nap (1 μm diamond suspension) provided by Struers. For the macro-scale microstructural evolution investigations, etched specimens were observed under Optical Microscope (OM) and Scanning Electron Microscope (SEM). The etching was carried out on both E-PBF and L-DED specimens using Kalling's No. 2 reagent (5 gm CuCl2, 100 ml HCl, 100 mm $CH_3CH_2OH$) for ~20 s for observations under the OM and FESEM for revealing the grain morphologies. An extra step of polishing with OPS-suspension on MD-Chem was carried out for specimens destined for EBSD analyses.

Olympus SZX7 stereo microscope (Magnification Range: 10x −50x) and ZEISS Axioskop 2 MAT optical microscope (Magnification Range: 50x−500x) were used to study the melt pool shapes obtained in the L-DED and E-PBF samples. The melt pool measurements were performed with the open-source image analysis software – Fiji (formerly ImageJ) on the optical micrographs. 3D laser scanning confocal microscope VK-X260k from Keyence Corp. was employed for capturing the large-scale micrograph of the 'POC' part shown in Fig. 1.

JEOL's JSM 7600 F field emission scanning electron microscope (FESEM) equipped with Oxford instrument's electron back-scattered diffraction (EBSD) detector was employed for the microstructural analysis of the 3d printed samples. The orientation mapping was carried out at an accelerating voltage of 20 kV and a probe current setting of 17 at an aperture setting of 1. The map settings were provided to the AZTEC software associated with the FESEM. The EBSD maps were utilized for crystallographic texture investigations and grain size analysis using the Aztec Crystal software. Fractured samples obtained post-tensioning of E-PBF fine- and coarse-powder samples were manually ground to ~50 μm in thickness. Φ3 mm disks were punched out from these specimens ~3 mm away from the fracture surface which was subsequently ground on the Gatan's 656-model dimple grinder and successively milled in the Gatan's 691-model precision ion polishing system at 3.5 kV and a milling angle range of 4−8°. These were analysed using the JEOL JEM-2010 TEM with a beam accelerating voltage of 200 kV.

**Machine learning framework.** Supervised ML66 via three-layer neural networks with five-fold validation was carried out on text-based data containing information about the PSD, thermal conductivity, and powder-bed packing density for a random set of granular feedstock – the framework of which has been presented in Fig. 4 of the main text. The PSD-specific thermophysical properties of the powder bed were computed by performing finite element analysis (FEA) based heat transfer simulations in the commercial software ABAQUS on a simulated particulate microstructure generated by Mote3D[35]. Twenty-six sets of data points containing the mean and deviation of distinct PSDs with their corresponding thermophysical properties obtained from the simulated powder beds were used to build (twenty data points) and validate (six data points) the neural network to estimate the powder-bed thermal conductivity (W m$^{-1}$K$^{-1}$) and packing density (%).

**Particulate powder bed simulations.** Numerical simulations for the powder-bed thermophysical properties for fine and coarse powders employed in the E-PBF process were carried out using the coupled Mote3D[35] and Abaqus®-FEA framework. Mote3D employs .m libraries compatible with the GNU Octave platform for the generation and random positioning of the spherical particles in a specified cubical domain. With the help of these libraries, we specified the edge length

of a cubical domain as 200 μm, the total number of particles, an intended mean and deviation of the PSD, and a particle overlap factor of 10% to ensure that there were sufficient contact regions for the conduction heat transfer simulations performed in the Abaqus®. After the execution of the program, the particulate microstructure was exported in voxel form to an Abaqus® compatible python file. By default, the Mote3D program gives the following choices of elements for voxel mesh generation: C3D8 (3D 8-node linear isoparametric element), C3D8R (R stands for reduced integration), C3D20 (3D 20-node quadratic isoparametric element) or C3D20R. For our case, we modified the code to generate elements DC3D20 which are specific for heat transfer applications. A total of 125,000 elements were generated in the simulated domain. Within the Abaqus® suite, constant thermophysical properties of SS316L – density (7950.0 kg m$^{-3}$), thermal conductivity (13.79 W m$^{-1}$K$^{-1}$), and specific heat capacity (470.0 J kg$^{-1}$K$^{-1}$), were applied to the voxel mesh. Heat transfer simulations in conduction mode were performed with a fixed temperature difference of 200 K between the top and bottom surfaces only.

**Discrete particle method (DPM) simulations.** DPM simulations for estimating the particle-size dependent flow velocities in the L-DED process were performed via 2D axis-symmetric modeling in COMSOL Multiphysics v 5.2 software where the flow of argon through the nozzles was modeled appropriately to the experimental conditions and average particle sizes were increased from 10 to 90 μm in 20 μm intervals and were assigned the thermophysical properties of SS316L. The determination of time of flight requires the estimation of particle velocity through the laser irradiation zone for a given particle size – which could be approximated with an exponential decay function as shown in the Supplementary Fig. 14. The time of flight was then calculated using the vertical travel distance of the powders in the illumination zone of 0.55 mm for a nozzle stand-off distance of 3.5 mm (Supplementary Fig. 16).

**Thermophysical properties of the sintered powder bed.** Since the E-PBF process takes place on a sintered powder bed, it was necessary to evaluate its thermophysical properties and their dependence on the feedstock PSD. Sintered powder-bed samples, from both fine and coarse powders, 5 mm × 5 mm × 1 mm in dimensions were carefully extracted from the region located beneath the base plate for measurements concerning thermal diffusivity, specific heat capacity, and packing density via laser flash apparatus (LFA), differential scanning calorimetry (DSC), and micro-computed tomography (μ-CT), respectively. NETZSCH's LFA 457 Microflash® was utilized for temperature-dependent thermal diffusivity measurements for temperatures ranging from room temperature (RT - 26 °C) to 850 °C. Along with the sintered powder bed, bulk specimens from EBM-printed fine- and coarse-powder samples were also prepared and tested under the same conditions. All of the samples were coated with graphite before the measurements. The measurements were performed in an N₂ environment for a laser operating voltage of 1538 V and a ramping rate of 2−5 °C min$^{-1}$. Mettler Toledo's DSC under the TOPEM® mode was employed for the specific heat capacity measurements of the sintered powder beds at RT and 450 °C. The measurements were performed in a N2 environment with a pulse height, width, and ramping rate of 1 K, 15−30 s, and 2 °C min$^{-1}$, respectively. The sample dimensions were further reduced to 3 mm × 3 mm in order to be accommodated in the DSC pan with a 4 mm diameter.

The 3D micro-CT analyses, results from which are shown in Fig. 4, were performed in a Bruker's Skyscan 1173 using a 130 kV X-ray source and a spot size of 5 μm. The raw data generated from the analyses were processed through the 3D.SUITE software provided with the Bruker system. The FESEM micrographs of the sintered powder beds were obtained using JEOL's JSM 7600 F and shown in Fig. 4.

**Mechanical testing.** Uniaxial tensile testing of E-PBF SS316L samples was carried out under a constant loading speed of 0.2 mm min$^{-1}$ using Shimadzu AGX 10 kN equipped with a non-contact digital video extensometer. Coupons were prepared from blocks of dimensions 30 mm × 30 mm × 25 mm. The gauge length of samples was 10 mm for the horizontal direction and 5 mm for vertical direction. The initial cross-sectional area for both types of samples was 3 mm$^2$. The strain rate was 3.3E−04 s$^{-1}$ for the horizontal specimens and 6.7E−04 s$^{-1}$ for vertical specimens. L-DED tensile samples were extracted from 20 mm × 20 mm × 20 mm cubical samples and were tested along the build direction. The gauge length was 5 mm, cross-sectional area ~1.5 mm$^2$, and subjected to a strain rate of 1E-03 s$^{-1}$. L-DED samples were tested using a Zwick-Roell 5 kN uniaxial tensile tester with a laser extensometer.

## Data availability
All data are available in the main text or the supplementary materials. Additional information can be obtained from the corresponding author upon request.

## Code availability
The G-codes used for DED fabrication of the 'NTU' block this work are available from the corresponding author upon request.

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

## Acknowledgements

S.B.T acknowledges the National Research Foundation, Prime Minister's Office, Singapore. U.R acknowledges the Structural Metal Alloys Program of the Agency for Science, Technology and Research of Singapore (A18B1b0061). X.P.T. acknowledges NUS Start-up grant (22-5928-A0001) and Singapore MOE Tier 1 grant (22-4902-A0001). The authors are thankful to J. Radhakrishnan, S. Huang, Y.J. Tan, and P. Kumar for the fruitful discussions and Y. Kok for the particle size distribution analysis.

## Author contributions

Conceptualization: S.C., X.P.T. Methodology: S.C., C.W., X.P.T. Investigation: S.C. Funding acquisition: S.B.T., U.R., X.P.T. Supervision: S.B.T., U.R., X.P.T. Writing – original draft: S.C. Writing – review & editing: U.R., X.P.T.

## Competing interests

The authors declare no competing interests.
