## [Peer Review File · Nature Communications]

Powder-size driven facile microstructural control in powder-fusion metal AM processesReviewers' comments:

Reviewer #1 (Remarks to the Author):

The authors show a large body of work on their study encompassing multiple AM build processes, simulation approaches, implementation of existing models, and mechanical testing. Despite this expansive study the authors fail to adequately backup their claims and explain their methodologies. The results as presented do not seem to fully line up with underlying theories and understanding of solidification and the AM process, and there are several sections included that do not appear to add significant value to the work. If the primary claim of the authors is in fact true, that by simply changing powder feedstock size we can control microstructure, a more straightforward, direct, and clear presentation of those results is required. The authors additionally need to provide a clear hypothesis of the origin of this phenomenon, which at the moment seems to hinge largely on melt pool shape, which should be dominated by processing parameters alone. Upon reviewing all of the main text and supplementary material, this reviewer finds the paper to on the whole consist of numerous specious claims with a generally lacking discussion section for such a bold claim.

More detailed comments follow below:

Line 58-59. Wide powder size distributions generally also help with flowability and spreading of the powder effectively in PBF processes.

Line 99 Caption on Figure 1 is quite long. Perhaps try putting some of this back into the main body of the manuscript? For example the explanation of why certain powder sizes lead to certain microstructures warrants a more comprehensive explanation in the text for those less familiar with classical solidification and welding theory.

Are there transition regions when changing powder feedstock during the DED process? Have you experimented with how quickly you can transition between microstructures within a single layer, or are you using novel tool path strategies to achieve this transition?

Given the low dilution regions in DED processes, relative to powder bed processes, do you have issues with varying surface roughness when switching powder streams in a given build layer? Are you keep powder mass flow constant between the two feedstock streams or adjusting some other parameter to mitigate this?

Line 138. Can you explain in more detail why powder size would affect the shape of the melt pool? Generally the melt pool is much larger than the particles (100s of particles or more melted at once) so should be controlled primarily through processing parameters.

Line 143. Again your caption is quite long and the reader may be better served by putting some of this in the main text. Is this the preferred format for this journal?

Can you please explain your grain size metric in Figure. 2 (a) in more detail? Why are you only showing results from parameter set 3 and 16, which appear to be outliers from the rest of the parameter set. What was special about these parameter sets that led to such large differences in grain size?

Particle size-preheat correlation. Can you explain your main conclusion from this more clearly? You are saying that fine powders heat up less than coarse powders before they hit the melt pool. Do you have a phenomenological explanation for this? Given your results as shown in Fig 3, which you state were all printed with the same build parameters (adjusting only powder federate to have constant mass flow), then the only things that are ostensibly different are the powder size distribution and possibly

the powder chemistry. I understand why different melt pool shapes can lead to different microstructures, but am not sure why powder size alone should affect melt pool shape so dramatically. If it is fully based on preheating of powders as they fly toward the melt pool, please state that explicitly, or please provide additional hypotheses on the origin of this phenomenon.

It appears from supplementary material that your bulk prints of DED were made using parameter set #3. Is this correct? Can you please note the parameter set used in the text for the reader and on Figure 2.a so they do not need to dig into the Supplementary section for this?

Why was machine learning used? Did you try something simpler like a multivariate regression? You only have a few variables you are looking at so not sure what the aim was with an ML model.

Can you please explain the contour plot in Figure 4 (b) more clearly? I do not understand what deviation you are referring to. Standard deviation? Are you assuming a gaussian distribution for particle size, log-normal, or something else if you are talking about a deviation?

You believe a 4% change in powder-bed density is responsible for the observed microstructural changes?

Figure 5. (a) how are you identifying melt pool boundaries in your overlays of the EB\SD maps?

Line 319 "explosive heterogeneous nucleation" what is meant by this? Please soften the language. Yes there is a critical undercooling at which nuclei in the melt become "activated" but this phenomenon is not an "explosion".

Line 1055: "given the absence of experimental data for SS316L" This is one of the most highly studied alloys in the literature, for both welding, casting, and AM. You should be able to get relevant thermophysical properties, even at least for 316 in order to calculate more accurate numbers for the parameters "a" and "n". The original work by Hunt was done with very different alloys (I believe Al-Cu or Al-Zn) than that of Gaumann in the late 90s (looking at superalloys and simplified the equation to be applicable for DED type welding) and the exponents changed significantly. These have a very large impact on the location of the CET on the processing diagram.

Please do not abbreviate cooling rate as "CR". It is confusing as this is technically calculated as the product of thermal gradient and interface velocity (leaving units of temperature/time and commonly written as $G_{\text{SEP}} \cdot R$ or $G \cdot V$), and simply writing out cooling rate each time does not sufficiently expand the text.

Line 364. Please provide more detail on "numerical simulations" performed to calculate G and R. This generally requires a framework modeling at least heat transfer and favorable mass transfer with high spatiotemporal resolution (micron/microsecond), which is not what Hunt's model entails.

Line 460: Now you are saying the large ductility is from TWIP in 316L? I believe this has been reported before but you show no evidence of twinning induced plasticity in you material, either in main figures or supplementary material. What evidence do you have of TWIP? The authors that have shown this attributed the phenomenon to nitrogen in the process gas during laser powder bed printing, but your samples were processed via electron beam melting (under vacuum). Conventional TWIP steels are generally high Mn content, which is believed to lower the SFE to enable glide for the twinning mechanism, but that is roughly an order of magnitude more Mn than what is in 316L.

Figure S1: please plot PSD on same scale.

Figure S5: There are two #5 single-tracks, I assume this is a typo

Reviewer #2 (Remarks to the Author):

The effects of powder size on grain structure and mechanical properties of the samples printed by directed energy deposition-laser beam and powder bed fusion - electron beam were studied in this work. The powder size induced grain size change was observed, which was attributed to the powder size induced thermal condition change.

It is well known that the change of the powder size will lead to change of thermal condition. The thermal condition change will cause grain structure change. In additive manufacturing industry, it is a common practice to optimize processing parameter for each particle size distribution to make printed part consistent. It is not clear what is the new scientific insight gained from this research.

The particle size induced change of grain size depends on the processing condition. Under certain condition, the grain size change is very small (e.g., parameter set # 12 in Fig. 2a). The range of thermal condition that can be tuned by using different powder sizes is limited. Tuning beam power and scan speed could achieve far more control of microstructure. In industrial production, the powder size for each additive manufacturing technology is fixed within a small range to make the process repeatable. The proposed method of using particle size to tune cooling condition has limited practical applications. For example, in directed energy deposition, large powder is preferred. Small powder could be blown away by gases, it is hard to control.

The particle size induced yield strength change reported in this paper is relatively small even though the particle size has been intentionally varied in a large range.

Grain size is only one aspect of the printed sample. Porosity plays more important role.

The words, PowderMAGIC, "shatters the existing consensus", "fascinating level", "extraordinary possibilities" do not add meaningful value to the paper.

Reviewer #3 (Remarks to the Author):

Authors have demonstrated an effect of input powder size on printed microstructure in 316L stainless steel. Two printing methods were used, direct energy deposition (DED) and electron beam powder bed fusion (EPBF). Using coarse powders, fine equiaxed grains were reported for both printing methods. The reason for the differences was attributed to heat transfer processes. Finally, detailed mechanical and microstructural characterization results were reported, with excellent mechanical properties obtained for the DED printing with coarse powder.

The work and description is comprehensive and clearly reported. The results are very useful to the field of AM, where effects of powder size on print outcomes are only beginning to be studied. Using powders of different sizes opens many new opportunities in engineering microstructure. The extent of the differences observed between the two powder sizes is striking.

I have one major comment for the authors, regarding the chemical composition of the stainless steel powder. In the methods section, the authors report that two batches of pre-alloyed stainless steel powder were used and they report a nominal composition. Given the importance of the differences in powder batch, authors should do the following:

- 1) Report the powder production process used (argon or nitrogen atomization)
- 2) Measure and report the composition of both the coarse and fine powder, stating the elements already listed
- 3) Measure and report nitrogen, oxygen, and aluminum

Subtle differences in composition may affect the build outcomes. In particular, total oxygen (and the presence or absence of aluminum) may lead to differences in non-metallic inclusions that could also impact the columnar-to-equiaxed transition. The total oxygen may differ because the amount of oxygen in powders can be a function of powder size. Composition measurement will provide additional support to the mechanism proposed by the authors.

Responses to the Reviewers' comments

The authors thank the reviewers' critical and insightful comments on this manuscript. We have taken the every single comment very seriously and thus conducted additional experiments and simulations as well as many changes in figures, main text and the supplementary material to better address the corresponding questions and comments. We highlight the changes **in red** in the revised manuscript. Please find our detailed responses **in blue** as follows:

Reviewer #1 (Remarks to the Author):

The authors show a large body of work on their study encompassing multiple AM build processes, simulation approaches, implementation of existing models, and mechanical testing. Despite this expansive study the authors fail to adequately backup their claims and explain their methodologies. The results as presented do not seem to fully line up with underlying theories and understanding of solidification and the AM process, and there are several sections included that do not appear to add significant value to the work. If the primary claim of the authors is in fact true, that by simply changing powder feedstock size we can control microstructure, a more straightforward, direct, and clear presentation of those results is required. The authors additionally need to provide a clear hypothesis of the origin of this phenomenon, which at the moment seems to hinge largely on melt pool shape, which should be dominated by processing parameters alone. Upon reviewing all of the main text and supplementary material, this reviewer finds the paper to on the whole consist of numerous specious claims with a generally lacking discussion section for such a bold claim.

Response: We thank the reviewer for the critical and constructive comments on the validity of our work. As mentioned in the beginning part of our manuscript, microstructure control in AM is promising yet demanding, but facile, cost-effective yet sustainable approaches of engineering grain microstructure are yet to be achieved. After many fruitful discussions with colleagues and peers in the metal AM community, we are quite confident that powder size effects should be exploited more deeply, and as we have explored, they indeed impose significant impact on the microstructure and mechanical performance in AMed metal alloys. In this work, through more than 3 years research experience and efforts, (as recognised by the reviewer, this is an “expensive study”), we have been able to report this work with full confidence. Fortunately, with the critical questions and comments raised by the reviewers, we have the chance to further improve our theory and make some tweaks in experiments and simulations in the revision. Particularly, we have provided a more straightforward, direct and clear presentation of the microstructural control results in the updated Figure 1 and 2, and highlighted the hypothesis of the origin of the powder-size driven microstructural changes in the text on Pg. 2 as “ ***We hypothesize that the sizes and velocity of distinct particles impinging the melt pool in a DED process and the collective thermophysical properties of the powder bed in a PBF process should impact the melt pool geometry – hence its solidification parameters***” and Figure 1. If possible, we would like to use a brief hypothesis or theory to clarify your question: flow dynamics and thermophysical properties of varied power sizes drive the changes in melt pool solidification behavior, which enables the grain microstructural control. All the detailed responses can be found below.

More detailed comments follow below:

1. Line 58-59. Wide powder size distributions generally also help with flowability and spreading of the powder effectively in PBF processes.

Response: The authors thank the reviewer for this comment. Following the reviewer's advice, the following statement with suited references has been added to the revised manuscript on Pg. 2.: *“Additionally, measurements of the rheological properties of powders for PBF processes show higher flowability and spreadability for wide PSDs – particularly benefitting from coarse powder particles^{21,22}”*. We hope the revision conveys the idea which reviewer originally intended.

2. Line 99 Caption on Figure 1 is quite long. Perhaps try putting some of this back into the main body of the manuscript? For example the explanation of why certain powder sizes lead to certain microstructures warrants a more comprehensive explanation in the text for those less familiar with classical solidification and welding theory.

Response: The authors thank the reviewer for highlighting the lengthiness of the figure caption. We have reduced the caption size significantly and have referred to the figure in the main text where we discuss the impact of powder sizes to the melt pool. Our lengthy caption from the original manuscript:

“PowderMAGIC approach for grain morphology and size control in 3D printed SS316L microstructure. a and b Schematics highlighting the PowderMAGIC triggered CET obtained in this work in PBF and DED fabricated SS316L microstructure. a Schematic show that granular feedstock with fine PSD induces the formation of shallow melt pool that favours epitaxial grain growth resulting in coarse elongated grains. b FG nearly equiaxed microstructure is observed for the coarse granular feedstock that occurs due to drastic transformation of the melt pool shape to a semi-circular one. c Sophisticated grain microstructure control achieved in this work via PowderMAGIC. The resulting proof-of-concept (POC) part showcases internal structure variation that is invisible when observed on a mirror finished surface but reveals the three letters ‘NTU’ acronym to ‘Nanyang Technological University’ – when etched. d Microstructural information of the POC part with the letter zone ‘NTU’ fabricated using the coarse powder feedstock contrary to the matrix zone fabricated using the fine particulate feedstock (Methods and Fig. S 2 in the Supplementary). The wider and deeper melt pool topologies observed in the letter zone result in growth of nearly equiaxed FG grains vs coarse columnar grains observed in the matrix zone. The associated scalebar is of length 5 mm. The crystallographic orientation map associated with the etched micrograph highlights the differences in grain sizes in the two distinct regions – separated by a white coloured boundary resembling a sinewave. The scalebar associated with the electron backscattered diffraction (EBSD) map is 500 μm . e Tensile coupons were machined and tested in horizontal direction from the N-section of the POC sample with an intention to highlight the distinct deformation mechanisms in the L (letter) and M (matrix) regions (demarcated by black dotted lines). The IPFz map of the gauge section showcases the fracture initiation at the M region with an associated map

showing significantly higher number of twin boundaries (in red) in L region composed of fine equiaxed microstructure. The scale bar for large scale EBSD maps of the tensile gauge section is 2 mm.”

Has now been reduced significantly in the revised manuscript as:

“PSD-driven grain morphology and size control in 3D-printed SS316L microstructure. a Particle preheat temperature variation with size and laser power. b Powder-size schematic highlighting the time of flight and temperature variation with the particle size. Colour of the particles corresponds to the temperature (blue to red depicting low to high temperatures, respectively). c The particle size preheat temperature variation obtained for 300 W laser power and the PSDs measured for fine and coarse powders. d Sophisticated grain microstructure control achieved in this work with a proof-of-concept (POC) part via particle-size dependent MPE approach. e Melt pool and microstructural information of the POC part. The associated scalebar is of length 5 mm. The scalebar associated with the electron backscattered diffraction (EBSD) map is 500 μm . f Results of the uniaxial tensile test showing distinguished response of the fine and coarse powder regions to deformation. The scale bar for large-scale EBSD maps of the tensile gauge section is 2 mm.”

As the reviewer would notice, aside from the revision in the caption required for the revised Figure 1, we have kept only necessary information regarding the figures and instead have added important information regarding the impact of powder sizes on the melt pool and microstructural evolution in the main text for on Pg. 3 in the Results section. We thank the reviewer for their recommendations and hope these corrections are to their expectations.

3. Are there transition regions when changing powder feedstock during the DED process? Have you experimented with how quickly you can transition between microstructures within a single layer, or are you using novel tool path strategies to achieve this transition?

Response: The authors appreciate this query. Yes, indeed we observed transition regions when changing the powder feedstock during the printing of proof-of-concept part shown in Figure 1 of the manuscript. For the implementation of the transition from fine to coarse powder and vice-versa in the DED system, we generated a G-code wherein, once the region intended for fine powder is finished, the turntable in the hopper with the fine powder stops and one in the hopper for coarse powder starts immediately without interruption the laser power and nozzle movements. However, as the powders are being fed to the nozzle assembly via a piping system, we observed an approximate time gap of ~ 3 seconds until the change in the turntable motion is reflected on the powder falling through the nozzle. Hence, when powder changes gradually from fine to coarse at the nozzle outlet in ~ 3 seconds. In this time, the nozzle moves a distance of ~ 50 mm at a scanning rate of 1000 mm/min. For our 20 mm long sample, this means 2.5 scan tracks before the powder changes completely from fine to coarse. As a result we observe a similar transition region of approximately 2 melt pools when the powder changes from fine to coarse when observed in the sectional view of the P-O-C sample. As we have noted that the fine powder flows faster than the coarse powder, the transition from coarse to fine powder is observed only after 1 – 1.5 melt tracks. Because the limitations exist with the powder delivery system, we did not experiment with the quicker transitioning from fine to coarse powder and vice-versa. However, we envision that for

slower scanning speed and higher carrier gas flow rate from the hoppers, it is quite possible to reduce the transition regions.

As a revision to Fig.1 we have now highlighted the transition regions aside from the distinct fine and coarse powder regions as shown below between yellow dashed lines. We hope this answers the reviewer's query and additionally thank them for this improvement to the manuscript.

4. Given the low dilution regions in DED processes, relative to powder bed processes, do you have issues with varying surface roughness when switching powder streams in a given build layer? Are you keep powder mass flow constant between the two feedstock streams or adjusting some other parameter to mitigate this?

Response: The authors are extremely thankful to the reviewer for their insightful queries posted in this comment. As the reviewer can note that the proof-of-concept part printed to showcase the site-specific microstructural control has the top surface printed using just fine powder, it was not possible for us to measure the variations in the surface roughness as that would require interruption of the build to a height at which there are distinct regions of fine and coarse powders (such as mid-height of NTU logo). However, for a qualitative understanding that we observed the single tracks of SS316L provided in the supplementary, for the same process parameter set of 300 W, 1000 mm/min, and 3.25 g/min (#3), the heights of the fine and coarse powder beads remained as 364.6 and 375.0 μm , respectively. Hence, when deposited in an overlapping condition, an approximate surface roughness of $\sim 10 \mu\text{m}$ will occur. However, due to the method with which the feedstock change occurs during the printing of the P-O-C part, the powders falling into the melt pool will gradually change from fine to coarse through what we believe an FC powder. Hence the melt and bead dimensions will be akin to those we observed using FC powder. This might be helpful in mitigating the surface distortions that might occur during powder change. It is also evident from the etched micrograph of P-O-C part in Fig. 1, the fine powder melt tracks are followed by around 2 - 3 transitional melt tracks to the coarse powder region.

To answer the second question in this comment, we are indeed keeping the mass flow rate constant as 3.25 g/min for the two feedstock streams for the fabrication of P-O-C part. We hope these responses suit the queries raised by the reviewer.

5. Line 138. Can you explain in more detail why powder size would affect the shape of the melt pool? Generally, the melt pool is much larger than the particles (100s of particles or more melted at once) so should be controlled primarily through processing parameters.

Response: The authors are extremely thankful for this question. As the impact of powder-size on the microstructure via variation in the melt pool forms the core of our hypothesis in this work, it is extremely important that we provide enough discussion on the same in the manuscript. Considering this we have revised our introduction section as well as the results

section substantially. We have added the following content to the results section on Pg. 3 where we discuss the role particle sizes play in their preheating as they traverse the laser illumination zone in the L-DED process:

“It is evident from Eq. (1) that the in-flight temperature rise of particles in a feedstock for the same process parameters and same material properties will depend upon their size and time of flight through the irradiation zone. The effect of increasing time of flight with particle size results in a complex variation in the particle preheat temperature (Fig. 1 a – c) as determined through discrete particle method (DPM) simulations (Methods and Section 6-Supplementary text).

Our investigations reveal several key insights on the compound effect of linearly increasing particle size and their exponentially decaying transit velocity on their preheat temperature in the powder-based L-DED process. We found out increasing particle size from 10 μm until $\sim 35 \mu\text{m}$, the preheat temperature falls significantly with extremely fine particles having very high temperatures (Fig. 1 a). Beyond the 35 μm particle size, the preheat temperature increases with the particle size until around 120 μm where the highest temperatures are predicted beyond which the impact of particle sizes supersedes the time of flight and the temperature falls. Hence, depending upon the sizes of particles in a PSD, the energy incident to the melt pool in form of preheated powders will vary and affect the melt pool formation. Also, with increasing laser power above 400 W, the particle preheat temperatures go beyond the evaporation temperature of 316L SS, this would consequently result in increasing evaporation of particles as they traverse through the laser and decreasing deposition of powders into the melt pool as evident from the decreasing deposition bead height and width for 500 W and 600 W single tracks (Fig. S 5 and Fig. S 15 in Supplementary text).

For 300 W laser power (Fig. 1 c), the fine PSD contains particles from region 1 where the temperatures are extremely high reaching the evaporation temperature for SS316L (~ 3000 K) hence the particles will be lost to evaporation whereas the coarse powder particles remain primarily in superheated state primarily in molten state. As the powder particles act as a secondary heat source to the melt – the power of which was determined for the fine and coarse powder feedstocks employed by us for a powder flow rate of 3.25 g/min as 85.2 W and 92.0 W, respectively (in See Section 4- Supplementary text). A higher preheating of the molten coarse powder particles results in higher incident energy to melt pool, which would affect the melt pool dimensions, their solidification parameters and the resulting microstructural evolution. ”

We hope that the reviewer would appreciate the addition of the new data and the results to the revised manuscript.

We would also like to provide an explanation to the reviewer here as from the query it is a bit unclear if it was meant specifically for L-DED or E-PBF process so here we will summarize the reasoning for both the processes:

1. In the DED process, the powders play a direct role in affecting the melt pool solidification parameters and resultant microstructure. As the powders are fed coaxially through the laser to the melt pool, they pass through the laser illumination region before hitting the melt surface. During this motion through the illumination zone, heating of the particles occur which has been shown to be affected in direct proportion to the time of flight and inversely to the particle size aside from the

parameters such as laser power, specific capacity, density et al as shown on Pg. 2 and 3 in the revised manuscript. These preheated particles incident to the melt pools act as a secondary power source and collectively heat up the melt pool. We showed through DPM simulations that the coarse powder particles moved slowly in the illumination zone in comparison to the fine powder and consequently heat up for a longer time. We calculated that for the particle size distribution and the powder flow rate of 3.25 g/min employed by us for the deposition of block samples, an equivalent energy/second input in from of preheated fine and coarse powders came out as 85.2 W and 92.0 W, respectively. These values are again dependent on the laser power and the powder flow rate aside from the material properties.

2. In the E-PBF process, the powders play a key role in governing the melt pool solidification parameters. The electron beam is first incident on a sintered powder bed whose thermophysical properties govern the heat transfer prior to the melt pool formation. During the solidification of the melt pool the heat dissipation – in absence of convection is through radiation and conduction from the powder bed and the deposited metal.

We hope these responses will be of liking to the reviewer and we appreciate these comments as the corrections have greatly benefited the manuscript.

6. Line 143. Again your caption is quite long and the reader may be better served by putting some of this in the main text. Is this the preferred format for this journal?

Response: The authors duly note this comment and have modified the caption accordingly. Our original intention was to provide enough information for the readers to not be disjointed from the figure. However, as the reviewer has pointed out, these turn out to be drabby. The caption for figure 2 from the original manuscript:
“Melt-pool architecture and grain morphology in DED printed single tracks of 316L SS. a Variation of a parameter that defines collective grain morphology (grain width and shape) with the process parameter sets employed for printing (Error! Reference source not found. in the Supplementary) – determined through EBSD mapping. Higher value of this parameter implies finer equiaxed grains. Points (i) and (ii) belong to the maps from coarse powdered feedstock as shown above the curve with point (iii) corresponding to the top region of the bead in (ii) with even finer grains. Point (iv) and (v) belong to the maps from fine powdered feedstocks. White dashed lines in the EBSD maps of the single tracks demarcate the melt pool interface. Columnar, equiaxed, and transition regions per visual inspections in the single-track beads have been demarcated using black coloured dashed lines. b Schematic of a deposited bead with relevant dimensions and nomenclature. h, d, and w – are notations for bead height, melt pool depth, and melt pool width, respectively. These 3 notations can together be reduced to two parameters Dilution = d/h , and Dimension = h/w . Higher Dilution implies deeper melt pool or smaller bead height and vice-versa. Higher value of Dimension implies taller or narrower bead and vice-versa. c, d, and e Variations in Dilution and Dimension with each of the three process parameters (power, scanning speed, and powder flow rate) obtained for the 17 beads out of 18 deposited (parameter set #1 does not have enough energy density to produce tangible beads) obtained for the two powder feedstocks. c Dilution/Dimension vs laser power. Dilution of beads deposited using coarse granulated feedstock is higher for power > 300 W.

Inverse behaviour is observed in Dimension vs power curve. The gap in the dilution and dimension for the two feedstocks increases gradually with power before stagnating after 500 W. d Dilution/Dimension vs laser scanning speed. Dilution of coarse powder higher than fine powder for the range of scanning speed of 500 – 3500 mm/min. Inverse behaviour observed in Dimension vs scanning speed curve. Gap in dilution for the two feedstock increases gradually. e Dilution/Dimension vs powder flow rate. Dilution in coarse powder beads higher than fine powder beads.”

Has been now been revised to:

“Melt-pool architecture and grain morphology in DED printed single tracks of 316L SS. a Variation in the grain widths and aspect ratios obtained from single-track depositions for process parameter set 2 – 18. b Schematic of a deposited bead with relevant dimensions and nomenclature. h , d , and w – are notations for bead height, melt pool depth, and melt pool width, respectively. c Dilution and Dimension vs laser power. d Dilution and Dimension vs laser scanning speed. e Dilution and Dimension vs powder flow rate.”

We have put major information to the discussion in the main text on Pg. 7 of the revised manuscript. We hope the revision is of liking to the reviewer.

7. Can you please explain your grain size metric in Figure. 2 (a) in more detail? Why are you only showing results from parameter set 3 and 16, which appear to be outliers from the rest of the parameter set. What was special about these parameter sets that led to such large differences in grain size?

Response: The authors appreciate this query. We chose this grain metric as a single parameter to show the transition of fine-equiaxed to coarse-columnar grain growth using the various 18 process parameter sets of the single track bead deposition. We used an inverse of the product of grain width and aspect ratio to visibly appeal to reader that the higher the metric the finer and more equiaxed the microstructure. However, upon revision of the manuscript we realized that this metric proves little purpose in advancing the understanding of the reader as it is too complex to correlate the same with the variations in the process parameters. Hence, we have significantly revised the figure 2 to showcase the variations in grain width and aspect ratio as independent quantities as shown below:

As seen in the new curve simple trends can be charted in grain sizes and aspect ratios with the processing parameters employed for the deposition of the single tracks which have been discussed in the results sections as well on Pg. 6 of the revised manuscript.

We highlighted the comparative EBSD maps for the fine and the coarse powder for parameter set #3 as this was the same parameter that was chosen for printing of the block samples. We may argue that it should not be an outlier as it follows the general trend from #2-#6 whereby

an increasing laser power increases the tendency to form similar microstructure in the fine and coarse powder samples as illustrated by the decreasing separation between the fine and coarse powder metrics until they overlap for parameter #6. We do agree with the reviewer that in the original manuscript #16 was clearly an outlier as from parameter set #13 - #18 we observe a general trend in the reduction of the grain metric for the coarse powder and inverse for the fine powder. For fine powders, parameter set #16 follows the trendline. To avoid any confusion to the reader, we have modified Figure 2 accordingly. Additionally, we have presented the comparative EBSD maps for the fine and the coarse powders for parameter sets #2-#18 in supplementary figures S5-S7 on Pages 35, 36, and 37.

8. Particle size-preheat correlation. Can you explain your main conclusion from this more clearly? You are saying that fine powders heat up less than coarse powders before they hit the melt pool. Do you have a phenomenological explanation for this? Given your results as shown in Fig 3, which you state were all printed with the same build parameters (adjusting only powder federate to have constant mass flow), then the only things that are ostensibly different are the powder size distribution and possibly the powder chemistry. I understand why different melt pool shapes can lead to different microstructures, but am not sure why powder size alone should affect melt pool shape so dramatically. If it is fully based on preheating of powders as they fly toward the melt pool, please state that explicitly, or please provide additional hypotheses on the origin of this phenomenon.

Response: The authors appreciate this query as this was one of the key factors which drove us to do significant revisions to our particle-size preheat correlation section. As the reviewer would know our earlier manuscript considered on two particle sizes – representatives of fine and coarse powders and determined the proportional differences in the preheat temperatures for the two feedstocks. However, we wondered as well that ideally finer powders – from the size point of view, should heat up faster than the coarse powders. And a given particle size distribution should have both fine and coarse powders and the rise in temperature of a particle should be independent of the PSD it was in. Hence, we tasked upon to provide a better phenomenological explanation for this phenomenon and we are glad to share that with the reader and the reviewer. As it has been detailed in the response to reviewer’s query #5 and in the main text of the revised manuscript on Pg. 2, we shall direct the reviewer to the same. But we would like to state that following reviewer’s recommendations we have added a clear hypothesis for the origin of this phenomenon on Pg. 2 as:

“We hypothesize that the sizes and velocity of distinct particles impinging the melt pool in a DED process and the collective thermophysical properties of the powder bed in a PBF process should impact the melt pool geometry – hence its solidification parameters”

We have also initiated the results section with the explanation on the particle size preheat correlation as that would form a precursor to the microstructural variations observed in the L-DED process which was **driven entirely by the particle size distribution**. We revised Figure 1 significantly as well to the following where the effect of the particle size on the preheat temperatures become far clearer. We hope this revision proves beneficial to the reader and thank again the reviewer for drawing our attention towards this problem.

9. It appears from supplementary material that your bulk prints of DED were made using parameter set #3. Is this correct? Can you please note the parameter set used in the text for the reader and on Figure 2.a so they do not need to dig into the Supplementary section for this?

Response: The authors appreciate this feedback from the reviewer. Yes, it is correct that the printing of the bulk sample fabrication for the L-DED process was done using the parameter set #3. Though we had provided the parameter set as (300 W, 1000 mm/min, and 3.25 g/min) in the methods of the original manuscript, we understand that upon highlighting the parameter set #3 will help in better connectivity of the research flow in the reader's mind in the way we originally intended. We thank the reviewer for providing this key comment. We have corrected the Figure 2a to reflect the parameter set employed for block sample fabrications.

10. Why was machine learning used? Did you try something simpler like a multivariate regression? You only have a few variables you are looking at so not sure what the aim was with an ML model.

Response: This is an excellent query by the reviewer. As the ML response curve shown in this work deals only with the variations in the thermal conductivity and powder bed packing density for varying mean and standard deviation of randomly generated PSDs – it must appear that the multi-variate line regression analysis would have been more apt for understanding the dependence of the two parameters with the PSD. However, we would like to clarify that the resulting ML-framework study detailed here is a subset of a much larger problem statement where we have envisioned to explore the variations in the number of powder particles in the simulation box, overlap ratios, simulation box sizes etc. We also envision exploring the particle shapes and their impact on the powder bed thermophysical properties. Another key factor i.e. the overlap ratio is the representative of the sintering effect produced by the beam and we believe a ML framework that can account for overlap ratio variations can prove beneficial to a range of PBF techniques – from L-PBF to E-PBF and even binder jet printing (BJP) process. We would like to highlight that the overlap ratio considered by us for our case was 38 % which was determined by comparing the powder bed density measurements obtained from the μ CT experiments to the simulated particulate microstructure for the PSDs corresponding to both fine and coarse powders. Hence, given that we had expertise in machine-learning techniques we decided to explore them with a point of expandability over line regression. We hope this answers the query raised by the reviewer and we hope they would appreciate that our ML response curve has been able to predict even the supposedly contradictory reports on powder-size microstructure dependence as highlighted by us on Pg. 9 and 10.

11. Can you please explain the contour plot in Figure 4 (b) more clearly? I do not understand what deviation you are referring to. Standard deviation? Are you assuming a gaussian distribution for particle size, log-normal, or something else if you are talking about a deviation?

Response: The authors are thankful to the reviewer for this query as this allowed us to improve our main text that involves the discussions about the Figure 4. We have revised the discussion prior to the figure on Pg. 11 as below and hope these answer the queries raised by

the reviewer as well:

“The contour plot, *with an assumption of normal size distribution in the particle size statistics*, indicates the variation of the relative powder bed density (from 25 to 75 %) with respect to the particle mean size and standard deviation and the lines show the variation of thermal conductivity in the intervals of 1, 2, 3, 4, and 5 W/mK.”

Response to the reviewer not added to the main text as details of the model are provided in the supplementary:

The particle size statistics were a generated dataset by the Mote3D opensource toolbox that was employed by us for generating virtual powder beds which were then virtually tested using Abaqus for the determination of thermal conductivity from a unidirectional heat conduction condition and powder bed density. At the time of this study, Mote3D only supports two types of particle size distribution metrics – normal distribution for distinct values of mean and standard deviation and monosize for a deviation value of zero. We thank the reviewer for raising this query.

12. You believe a 4% change in powder-bed density is responsible for the observed microstructural changes?

Response: The authors thank the reviewer for this query. Of course, it is indeed true that from micro-CT measurements we could measure the density of fine powder sintered powder bed as 77 % as opposed to 81 % measured for the coarse sintered powder bed. However, we do not believe nor that we claim that only the slightly higher powder bed density is the reasoning behind the out of ordinary equiaxed microstructure obtained the EBM printed coarse powder samples. It should be noted that we also report a higher thermal conductivity for the coarse sintered powder bed using the laser flash apparatus. Both of these factors together contribute to the increased heat transfer from the coarse sintered powder bed. But our main contribution to the microstructural variation comes from us “tricking” the Arcam

EBM's automation software. To reiterate from the supplementary text, even though the technology behind the automation software is a black box, we believe it works by measuring the temperature difference between the first layer of sintered powder bed for a fix material and employs changes to beam parameters to compensate for more heat loss or lack-there-of accordingly. Our hypothesis was that if we optimized the processing parameters for SS316L using the fine powder and use the same parameters for printing coarse powder the automation software might be tricked into "believing" it is printing a material with better thermal conductivity due to improved heat transfer from the powder-bed. This will lead it to control the beam scanning speed with an intention to generate more heat in the powder bed as its task is to maintain the build temperature specified for the processing. But as the base material is a solid block or deposited layers of SS316L have same material properties – as shown in the LFA measurements, the coarse powder deposited layers will be subjected to a higher equivalent energy density and a suitable melt pool shape, solidification parameters and microstructural evolution will accompany. We were in fact proven correct that the EBM system indeed changed the process parameters substantially without a human intervention for the coarse powders and resulted in the equiaxed microstructure as observed by us. We hope this explanation provides an adequate response to the query raise by the reviewer. We have also revised the text accordingly to convey this message clearly and avoid further confusions to the readers.

13. *Figure 5. (a) how are you identifying melt pool boundaries in your overlays of the EB\SD maps?*

Response: The melt pool boundaries for the coarse powder samples was identified with the help of the band contrast map as shown below originally associated with the IPF map presented in Figure 5 (a):

As can be seen in the figure, the coarse powder melt pool boundaries could be identified from the faint lines present in the band contrast maps. However, as the same was not identifiable from the BC maps of the fine powder samples, the melt pool boundaries for fine powder samples were observed roughly around the same region using etched micrographs, as shown below, and overlaid to the EBSD map.

We hope our response is apt to the query raised by the reviewer.

14. Line 319 “explosive heterogeneous nucleation” what is meant by this? Please soften the language. Yes there is a critical undercooling at which nuclei in the melt become “activated” but this phenomenon is not an “explosion”.

Response: The authors sincerely thank the reviewer for highlighting the shortcomings such as these in the original manuscript. We realise use of the term explosive does seem to overarch the nucleation which occurs hence we have removed it from the revised manuscript.

15. Line 1055: “given the absence of experimental data for SS316L” This is one of the most highly studied alloys in the literature, for both welding, casting, and AM. You should be able to get relevant thermophysical properties, even at least for 316 in order to calculate more accurate numbers for the parameters “a” and “n”. The original work by Hunt was done with very different alloys (I believe Al-Cu or Al-Zn) than that of Gaumann in the late 90s (looking at superalloys and simplified the equation to be applicable for DED type welding) and the exponents changed significantly. These have a very large impact on the location of the CET on the processing diagram.

Response: The authors are sincerely thankful to the reviewer for raising their concern regarding the use of a generic ‘a’ and ‘n’ value that were originally used for Ni-based superalloys. It was indeed an oversight on our part, and we have rectified our mistake as following. We determined these parameters by power law curve fitting of the solidification velocity – undercooling variation curve for SS316L provided in the paper “Lin, Xin, et al. "Columnar to equiaxed transition during alloy solidification." Science in China Series E: Technological Sciences 46 (2003): 475-489.” The parameters ‘n’ and ‘a’ were determined respectively as 2.94 and $5.87 \times 10^4 \text{ K}^{2.94} \cdot \text{m}^{-1} \cdot \text{s}$ and were employed for plotting of the new CET curves as depicted in Figure 5. The authors again thank the reviewer as provision of these data will be beneficial in accessing the quantitative information to the future researchers working on SS316L. Following is the original versus revised CET curve.

16. Please do not abbreviate cooling rate as “CR”. It is confusing as this is technically calculated as the product of thermal gradient and interface velocity (leaving units of temperature/time and commonly written as $G \cdot R$ or $G \cdot V$), and simply writing out cooling rate each time does not sufficiently expand the text.

Response: The authors are thankful to the reviewer for pointing out this correction. We have accordingly mitigated this mistake and have now used the correct terminology of cooling rate in the revised text.

17. Line 364. Please provide more detail on “numerical simulations” performed to calculate G and R . This generally requires a framework modeling at least heat transfer and favorable mass transfer with high spatiotemporal resolution (micron/microsecond), which is not what Hunt’s model entails.

Response: We thank the reviewer for this query. To address the concerns raised we have now added to the details of the numerical modelling which were already provided in the section 6 of the supplementary material added to the original manuscript. We hope these corrections added on Pg. 28 and Pg. 29 are on par with the expectations of the reviewer:

“For the fine and coarse powder E-PBF process simulations, a time step of 0.1 μs was employed for a total simulation time of 5 ms. An initial mesh was generated using predefined metrics of ‘Fine’ and calibrated for fluid dynamics in the COMSOL Multiphysics software. For this setting the element size range was from 5 – 175 μm . Further, the region below the top surface – where the simulated heat flux was incident, had a set of 40 boundary layers having a stretching factor of 1.1 with the smallest boundary layer thickness of 3 μm just below the surface. Adaptive mesh refinement was enabled with the time dependent linear (PARDISO) solver employed for the heat transfer simulations. A smallest mesh size of 1.5

μm was obtained at the end for coarse powder simulations while 2.8 μm was obtained in the end for fine powder simulations.”

“A time step of 1 μs was employed for a total simulation time of 50 ms for both the coarse and fine powder simulations for the L-DED process. The initial meshing parameters were the same as that for the E-PBF simulations except that the adaptive meshing resulted in the smallest element size of 10.8 μm for both fine and coarse powder simulations. The thermal history and thermal gradient values were determined at a point in the middle of the melt pool – 200 μm below the free surface.”

We wish to clarify that we did not perform either mass or fluid transfer simulations given their computationally intensive nature and instead focussed only on the heat transfer physics.

18. Line 460: Now you are saying the large ductility is from TWIP in 316L? I believe this has been reported before but you show no evidence of twinning induced plasticity in your material, either in main figures or supplementary material. What evidence do you have of TWIP? The authors that have shown this attributed the phenomenon to nitrogen in the process gas during laser powder bed printing, but your samples were processed via electron beam melting (under vacuum). Conventional TWIP steels are generally high Mn content, which is believed to lower the SFE to enable glide for the twinning mechanism, but that is roughly an order of magnitude more Mn than what is in 316L.

Response: We would like to state that the reviewer indeed raises a good point however, we did provide the evidences of deformation induced twinning obtained in the coarse powder samples. The authors would like to direct the reviewer’s attention to Figure 6 c- f of the original and revised manuscript where we showcase with the help of EBSD (detection of special boundaries) and TEM image evidence of the twins formed in the samples ~ 3 mm away from the fracture surface. It is indeed true that the higher Mn content might reduce the SFE, however, it has been established for a while that for fcc materials such as SS316L, SFE only has an indirect influence on twinning stress and instead the dislocation density and homogeneous slip length are the most relevant microstructural features that govern the twinning stress in a polycrystal (“Influence of Grain Size and Stacking-Fault Energy on Deformation Twinning in Fcc Metals. Ehab el-danaf, Surya R. Kalidindi, and Roger D. Doherty, j.a.p, 1999”). As this study was performed on conventionally manufactured and recrystallized SS316L, we would also like to refer to several earlier researches on SLM printed purely austenite SS316L which we believe are more relevant to our work where argon environment was employed during fabrication. Sun et al (reference #12 in our revised manuscript) reported significant twinning induced plasticity in <110> textured SLM printed SS316L as opposed to <001> textured SS316L. Wang et al. (Crystallographic- orientation-dependent tensile behaviours of stainless steel 316L fabricated by laser powder bed fusion) fabricated and tested strongly textured SLM deposited SS316L along the <001>, <110>, and <111> directions and report twinning induced plasticity in both <110> and <111> oriented grains as opposed to <001> grains. In their own conclusions:

“The crystallographic-orientation-dependant propensity towards deformation twinning was attributed to the different effective stacking fault energies resulted from different resolved shear stresses applied on the partial dislocation pairs in different crystallographic orientations. The critical stresses for deformation twinning (σ_{TW}) of the<110>and<111>orientations were

close to their flow stresses at yielding, while the σ_{TW} of the $\langle 100 \rangle$ orientation was much higher than its flow stress level.”

We hope our explanation proves satisfactory to the reviewer.

19. Figure S1: please plot PSD on same scale.

Response: The authors thankful for this diligent observation. The PSD for fine and coarse powders have been revised in Relative volume fraction in the same scale. The image S1 with the revised PSD curves in (e) and (f) is shown below:

Fig. S 1: Powdered feedstocks employed with the PowderMAGIC approach for L-DED and E-PBF processes. Scanning electron micrographs obtained for (A) fine and (B) coarse

SS316L powders (C) Malvern Panalytical Mastersizer 3000 with Aero S adapter (D) Principle of Mie scattering^{1,2} employed by the mastersizer for particle size determination. Powder-size distribution obtained from the Mastersizer with cumulative volume fraction for (E) fine and (F) coarse powders.

We hope this correction is on par with the reviewer's expectation.

20. *Figure S5: There are two #5 single-tracks, I assume this is a typo*

Response: The authors are sincerely thankful to the reviewer for pointing out this mistake. It was indeed a typo and we have corrected the second #5 to #6 which should have been the case to begin with.

Reviewer #2 (Remarks to the Author):

The effects of powder size on grain structure and mechanical properties of the samples printed by directed energy deposition-laser beam and powder bed fusion - electron beam were studied in this work. The powder size induced grain size change was observed, which was attributed to the powder size induced thermal condition change.

1. It is well known that the change of the powder size will lead to change of thermal condition. The thermal condition change will cause grain structure change. In additive manufacturing industry, it is a common practice to optimize processing parameter for each particle size distribution to make printed part consistent. It is not clear what is the new scientific insight gained from this research.

Response: We thank the reviewer for his/her professional understanding on the topic. To our knowledge, this is the first work that systematically explores the powder-size driven microstructural control in both the powder-blown directed energy deposition and the powder bed fusion processes. The new scientific insight gained from this research lies in the solid experimental proof of powder size effects (e.g. thermophysical properties and melt pool morphology), and sound analytical analysis and numerical simulations (e.g. powder flow dynamics, powder preheating, and melt pool solidification parameters). We understand that our original manuscript fell short in highlighting the scientific insights it was bringing to the AM community. Hence to address this, we have done major revisions and have revamped the manuscript to clearly present the insights. We hope the reviewer would give us another chance by reviewing this revised version again. We welcome any feedback with complete enthusiasm. Here, we have listed a few of the insights for better clarity:

1. Particle size-driven preheating of powders: We are confident that in this revised version of the manuscript, we have presented results on particle size preheat correlation like never before. As the reviewer is aware, earlier works have employed the preheat equation for powder particles in the L-DED processes that we have given in form of Eq. 1 on Pg. 2 of the revised manuscript. This analytical equation correlates the preheating of the powder particles directly to the time of flight of the particles and inversely to their sizes – among other processing parameters such as laser power, material properties etc. With resounding success this equation has been employed for estimating the extra energy incident to a L-DED melt pool in form of preheated powders – particularly useful for numerical simulations etc. However, to the best of our knowledge, often a single particle size and single time of flight have been employed that have shown to result in a single value of preheat temperature which has often been assumed to represent the entire powder flow. We have shown, by using the DPM simulations and analytical correlations that with the conditions employed by us – which resembled a typical DED process, the particle flow speeds in the illumination zone showed an exponential decay with increasing particle sizes (which are shown in the Fig. S14 of the supplementary text). Consequently, the resultant preheat temperature variation with particle size is a unique curve which to the best of our knowledge, has been reported for the very first time to the AM community. As seen in the revised Figure 1 of our revised manuscript, the preheat temperature values were extremely high for extremely fine particles, as the particle sizes increased, the preheat temperatures reduced until $\sim 35 \mu\text{m}$, when the valley of the curve is reached, beyond

which the effect of continually increasing time of flight supersedes the effect the increasing particle size had on the preheat temperature. As we observed in the supplementary figure S 14 in our revised manuscript that the time of flight begins to reach a plateau beyond 120 μm particle size, a peak in the preheat temperature curve is reached beyond which the increasing particle size governs the evolution of the curve. We believe this is a finding that can be beneficial to the AM community, as the sizes of the powder particles in their feedstock PSD would govern the average preheat temperature of the powders collectively as they pass through the laser beam to the melt pool.

2. Effect of laser power on the particle preheat temperature and bead/melt pool dimensions: To date researchers in the AM community have agreed to a consensus on the shape of the melt pool to the microstructural evolutions within them. It is now a wide established understanding that a convection melt pool (semicircular shape) would result in a primarily diffuse microstructure and vice-versa. However estimating the melt pool and the bead shape from the L-DED processing parameters is still far from understanding. We show from Fig. 1a that with increasing laser power, the variation in the preheat temperature with particle size increases as evident from the more defined valleys and peaks with higher power. Due to this for low power such as 200 W, the effect of particle size on the melt pool shape and microstructural evolution is negligible – evident from the overlapping grain size metric in Fig. 2a. The difference between the same increases with increasing laser power but for 600 W the difference is mitigated again as a majority of the powders are heated up to temperatures beyond the evaporation temperature of SS316L (~ 3000 K) fine and coarse powder alike. This also results in a sudden drop in the bead height for 600 W laser power as shown in Fig S15.
3. Site-specific microstructural variation in DED process: We are the very first to showcase variation in the microstructural hierarchical simply by changing the powder feedstock size in-situ during deposition of L-DED process – for the same material. We have shown the distinct microstructure that had imprinted “NTU” logo where the letter we printed with the coarse powders and the matrix with the fine powders – which were only visible during etching. Not only in the grain sizes and the melt pool dimensions we also observe differences in the dislocation densities of the coarse powder and fine powder deposited regions of this sample as seen below in the GND map obtained from the region having the letter “T”. Nowadays, hydrogen embrittlement of steels is gaining significant traction due to their obvious candidacy in hydrogen storage and transportation facilities. The role of dislocation densities in assisting or limiting hydrogen embrittlement is still under research but be as it may we present two regions of steel which have distinguished dislocation densities for similar YS and UTS. We hope reviewer would appreciate this application specific advancement our work can bring to this important field.

Figure. EBSD KAM map showing the varying GND density in the site-specific “T” shaped microstructure obtained in L-DED process with fine and coarse powder feedstocks.

2. *The particle size induced change of grain size depends on the processing condition. Under certain condition, the grain size change is very small (e.g., parameter set # 12 in Fig. 2a). The range of thermal condition that can be tuned by using different powder sizes is limited. Tuning beam power and scan speed could achieve far more control of microstructure. In industrial production, the powder size for each additive manufacturing technology is fixed within a small range to make the process repeatable. The proposed method of using particle size to tune cooling condition has limited practical applications. For example, in directed energy deposition, large powder is preferred. Small powder could be blown away by gases, it is hard to control.*

Response: We are thankful to the reviewer for the critical thinking. As they would notice we have noted outrightly in our introduction that process parameter variations are indeed the most prominent method for producing customizable microstructures. In fact, aside from the proposed powder size-driven approach, varying process parameter is the only way to achieve a two-way microstructure control from columnar to equiaxed transition and vice versa. We wish to again state that our work is in no way to undermine the state-of-art of process parameter variations. However, the current usage of specific powder size range for the specific AM process seems to be a stereotype. We may argue that the actual suitable powder sizes can be more flexible than expected for both DED and PBF processes, namely ~ 15 to 150 μm . For instance, the powder sizes recommended by the manufacturers are never specific to the material in question but to their machines. For instance, the recommended particle sizes for the printing of SS316L, In718, Ti6Al4V etc. using ARCAM EBM system we used is 45 – 105 μm . However, we can argue that when powder beds form the basis where an electron beam scans during the process, what advantage these three materials, with such different thermophysical properties, provide to have similar particle sizes. Another point for L-DED process can be made that as seen in the power and size dependent preheating curves in Fig.1 of our manuscripts, the temperatures above 3000 K may not be suitable for the SS316L fine particles but might just be what is needed for melting and printing of refractory metals such as W, Nb etc. where the melting temperatures reach this ceiling. Hence, printing of such hard, refractory metals using the DED process might be more facile using finer powder sizes. As seen in our ML results (Figure 4), the variable ranges in powder packing density are wide and

distinct. Thus, in terms of our powder size effect theory or hypothesis, the range of thermal condition is highly tuneable. Meanwhile, it is well established tuning beam power and scan speed could control microstructure but it is worth noting that for many metal alloys such as Ni-based superalloys, high entropy alloys, etc. that has limited process windows, the power of tuning process parameters for microstructural control is much limited. As per our AM experience, coarse powders are indeed preferred for DED process for the ease of handling. But it does not mean that fine powders would not be able to be used with low recycling rates. Regarding the reviewer's query that for parameter #12, the impact of powder size is not that significant. It is indeed an astute observation – however, we wish to point out that parameter sets #7 – 12 were printed with increasing scanning speeds gradually from 500 – 3500 mm/min. As the scanning speeds increases, the powder catchment efficiency suffers greatly. In fact, slower moving coarse powder particles reach the fast moving and fast solidifying melt pool far lower than the fine powders. This is the reason why a bead height of 650 μm obtained at 500 mm/min scanning speed is reduced greatly to $\sim 100 \mu\text{m}$ for 3500 mm/min as seen in Fig S15 of supplementary information. This also affirms our another point that at different process parameters such as this high scanning rate, the difference in grain size may be great in comparison to 500 mm/min deposition as seen in Fig. 2a, but the deposition efficiency for such a parameter reduces greatly by more than 6 times thus making those process parameters inefficient in producing bulk samples. Through our study we have attempted to provide a new dimension to tunability of the AM microstructure.

3. The particle size induced yield strength change reported in this paper is relatively small even though the particle size has been intentionally varied in a large range.

Response: The authors appreciate this comment by the reviewer. We do agree with this point however we believe that the minimum difference in the yield strength values observed are likely due to the following reasons:

1. It is being discovered that aside from the grain sizes, factors such as crystallographic texture, dislocation density, PDAS, grain boundaries and sub-grain boundary segregation also play important roles in determining the yield strength. And their roles in improving the yield strength are believed to be synergistic and anisotropic. For instance, in our Fine powder E-PBF printed samples, the yield strength obtained along the direction of the columnar grain was found to be higher than that perpendicular to it (Fine-V vs Fine-H, respectively).
2. In the DED samples, for a coarser grain size obtained in the fine powder samples, there are visible sub-grain boundaries present which is not the case for the DED printed coarse powder samples – due to the non-equilibrium solidification mode in the latter. We believe this is the reason for same YS values for the two powder samples

We would like to highlight however that our fine-equiaxed grains indeed improve the isotropy in the response and result in a significant improvement in ductility. In fact, as we have provided in the manuscript the UTS and ϵ_f combination obtained by us in the DED samples is the best reported for 3D-printed SS316L. As this is for as-printed state only without any costly and tedious post processing, our proposed microstructural engineering approach has demonstrated the promising outcome.

4. Grain size is only one aspect of the printed sample. Porosity plays more important role.

Response: This is a great point made by the reviewer. It is indeed true that porosity plays a significant role when determining the overall mechanical response of an additively manufactured alloy. This is the reason for us to optimize the processing parameters for both the DED and E-PBF process to achieve as close to fully dense samples as possible. Though we still observe a few rounded pores in the E-PBF fine and coarse powder samples (Fig. 5a in the manuscript), which primarily originate from the powder due to the gas-atomization powder production process and get entrapped in the samples due to the fast scanning speeds of the electron beam, our DED samples were completely devoid of the same as evident from the large-scale micrographs shown in Fig. 3. Given the powder production process was the same for the two batches of the powder, we did not investigate the impact porosity had on the mechanical response but indeed it will be of significant interest to explore this for future studies.

5. *The words, PowderMAGIC, "shatters the existing consensus", "fascinating level", "extraordinary possibilities" do not add meaningful value to the paper.*

Response: We truly appreciate the reviewer's feedback as it made us realize the use of such terminology in the original manuscript end up lacking specificity. As a result, we have made significant revisions to the manuscript and have rid it entirely of such jargons. We hope the revised version appears more grounded and focused to the reviewer as we have intended.

Reviewer #3 (Remarks to the Author):

Authors have demonstrated an effect of input powder size on printed microstructure in 316L stainless steel. Two printing methods were used, direct energy deposition (DED) and electron beam powder bed fusion (EPBF). Using coarse powders, fine equiaxed grains were reported for both printing methods. The reason for the differences was attributed to heat transfer processes. Finally, detailed mechanical and microstructural characterization results were reported, with excellent mechanical properties obtained for the DED printing with coarse powder.

The work and description is comprehensive and clearly reported. The results are very useful to the field of AM, where effects of powder size on print outcomes are only beginning to be studied. Using powders of different sizes opens many new opportunities in engineering microstructure. The extent of the differences observed between the two powder sizes is striking.

Response: The authors are incredibly grateful for the reviewer's positive and insightful feedback on their work. The authors especially appreciate the recognition of the significant impacts of powder size on printed microstructure in 316L stainless steel by the reviewer. We agree with the nascent stage of understanding of the powder size-microstructure-mechanical response correlations in the field of additive manufacturing. Our sincere thanks to you for recognising the probable impact of our work to the field.

1. I have one major comment for the authors, regarding the chemical composition of the stainless steel powder. In the methods section, the authors report that two batches of prealloyed stainless steel powder were used and they report a nominal composition.

Response: The authors truly appreciate this feedback as this helped us in strengthening our argument that the differences in the fine and coarse powder microstructures was from the powder sizes only. We have done the due diligence to address the reviewer's comments and enclosed are our responses to the specific comments.

Given the importance of the differences in powder batch, authors should do the following:

1) Report the powder production process used (argon or nitrogen atomization)

Response: We thank the reviewer for this query. Upon confirmation with the powder supplier, we report the following the manuscript:

The cast billets of the SS316L were melted in vacuum and atomized using argon gas for both fine and coarse powders.

2) Measure and report the composition of both the coarse and fine powder, stating the elements already listed

Response: As requested by the reviewer, we employed ICP-OES for the measurement of the elemental composition of the fine and coarse powders. The measurements are tabulated in the revised manuscript in the Table S5.

3) Measure and report nitrogen, oxygen, and aluminum

Response: As suggested by the reviewer, we employed ICP-OES (Agilent 720 series) for the measurement of Al composition of the fine and coarse powders and employed combustion method for the measurements of carbon by ELTRA CS800), hydrogen, nitrogen and oxygen by ELTRA ONH2000. The measurements are tabulated in the revised manuscript in the Table S5.

	Fe	Cr	Ni	Mo	Mn	Si	Al	C	S	O	N	H
Fine powder	Bal.	15.30	10.50	1.98	1.04	0.31	0.06	0.025	0.013	0.093	0.088	0.006
Coarse powder	Bal.	16.52	10.76	2.22	0.33	0.41	0.06	0.016	0.012	0.072	0.068	0.009

Subtle differences in composition may affect the build outcomes. In particular, total oxygen (and the presence or absence of aluminum) may lead to differences in non-metallic inclusions that could also impact the columnar-to-equiaxed transition. The total oxygen may differ because the amount of oxygen in powders can be a function of powder size. Composition measurement will provide additional support to the mechanism proposed by the authors.

Response: The authors are thankful to the reviewer for their insightful comments. As we can see from the compositional analyses, that the elemental compositions of the fine and coarse powders were within the nominal compositions stated by the manufacturer for SS316L hence, we can confidently eliminate the impact of compositions to the CET observed by us in our powder size driven microstructural control for fine and coarse powders. As we notice the Al wt. % was measured to be exactly the same in the two powders with the O wt. % even lesser in coarse powders we can confirm that the powder sizes are the driving factor in the microstructure change that was driven by the formation of different melt pool sizes for the fine, FC, and coarse powders. We are however sincerely thankful to the reviewer for recommending the measurement in their comment as this helped us in reaffirming our hypothesis and theory.

REVIEWERS' COMMENTS

Reviewer #1 (Remarks to the Author):

The reviewer appreciates the fine attention to detail taken by the authors in responding to my original set of inquiries, and I am overall satisfied with the current state of the article. I believe the magnitude of the impact of powder sizes is more clearly presented, and sincerely appreciate the additional analyses provided to help substantiate the authors' claims. I do agree that this is an interesting and exciting approach to enhance microstructural control in AM processes, and appreciate the thoughtfulness and depth of the authors in revising their manuscript.

Reviewer #2 (Remarks to the Author):

The authors added more data to support the claims in the paper. However, the novelty and potential impact of the paper has not changed. I am not convinced that this paper is publishable in Nature Communications.

It is well-known that the change of particle size could cause changes in microstructure and properties of the printed part. This is an important reason for part quality uncertainty. It is a common practice in industry to develop processing parameters for different batches of powders even with subtle change of the powder size.

It is claimed in the abstract that "Engineering the columnar-to-equiaxed transition during rapid solidification of the additive manufacturing process is crucial for the technological advancement but it remains formidable yet." Many solutions have been reported to achieve equiaxed grains in the printed parts: (1) tuning heat power and scan speed, (2) tuning preheating temperature, (3) tuning scan pattern, (4) tuning alloy composition, (5) adding ceramic particles, (6) tuning thermal conditions in the heated affected zone to promote recrystallization, (7) beam shaping, (8) using ultrasound. Many of them are much simpler and easier to implement than changing powder size. The usage of the word "formidable" does not add value to the paper.

Reviewer #3 (Remarks to the Author):

Authors have reported chemical composition analyses. While there are some differences, they are relatively small. This alone does not prove that there are no effects of composition, however the fine powder containing higher interstitials results in coarser grains, suggesting some other effect dominates.

Reviewer #1 (Remarks to the Author):

The reviewer appreciates the fine attention to detail taken by the authors in responding to my original set of inquiries, and I am overall satisfied with the current state of the article. I believe the magnitude of the impact of powder sizes is more clearly presented, and sincerely appreciate the additional analyses provided to help substantiate the authors' claims. I do agree that this is an interesting and exciting approach to enhance microstructural control in AM processes, and appreciate the thoughtfulness and depth of the authors in revising their manuscript.

Response: The authors thank the reviewer for appreciating our efforts in performing the revisions to the manuscript in response to their comments to the first draft. It was truly our privilege that we got such detailed comments from the reviewer. Their queries and corrections drove us in rethinking the impact of our work and performing significant revisions to the manuscript. We are grateful that the reviewer identified our efforts and recognizes the advancement our powder size driven technique brings to microstructural evolution control in additive manufacturing.

Reviewer #2 (Remarks to the Author):

The authors added more data to support the claims in the paper. However, the novelty and potential impact of the paper has not changed. I am not convinced that this paper is publishable in Nature Communications.

Response: The authors are thankful to the reviewer for providing their unbiased comments to the first and the revised manuscript. We however would like to sincerely disagree with the reviewer regarding their comment on novelty as well as potential impact of our work. In our previous response, we provided detailed accounts on the same enlisting three major points on the impact and novelty in the revised manuscript which are reiterated here:

1. Particle size-driven preheating of powders
2. Effect of laser power on particle preheat temperature and bead/melt pool dimensions
3. Site-specific microstructural variation in DED process

It is well-known that the change of particle size could cause changes in microstructure and properties of the printed part. This is an important reason for part quality uncertainty. It is a common practice in industry to developing processing parameters for different batches of powders even with subtle change of the powder size.

Response: We are thankful to the reviewer for their comments. We believe that with their expertise on the subject matter, it might be obvious to the reviewer that the particle sizes affect the microstructure. However, we did not find any such detailed description or correlations in the literature. We can confirm that to date there is a substantial research gap on the effect of powder size on the microstructure in the L-DED process. Whereas, in the PBF community, a consensus exists that the fine powder results in improved mechanical response. In fact, in the case-studies we provided in Figure 4 of the main text and in the associated discussions we present the reasoning behind this constrained point of view. We do agree with the reviewer that it

is indeed a common practice in the industry to vary the process parameters for different powder sizes with an intention to achieve uniformity in the mechanical response to reduce quality uncertainty. However, our belief is that purpose of academic research is to push the boundaries of state-of-the-art that later can be adapted as new industrial standards.

It is claimed in the abstract that "Engineering the columnar-to-equiaxed transition during rapid solidification of the additive manufacturing process is crucial for the technological advancement but it remains formidable yet." Many solutions have been reported to achieve equiaxed grains in the printed parts: (1) tuning heat power and scan speed, (2) tuning preheating temperature, (3) tuning scan pattern, (4) tuning alloy composition, (5) adding ceramic particles, (6) tuning thermal conditions in the heated affected zone to promote recrystallization, (7) beam shaping, (8) using ultrasound. Many of them are much simpler and easier to implement than changing powder size. The usage of the word "formidable" does not add value to the paper.

Response: We thank the reviewer for their comment regarding the statement in the abstract. We agree that the word formidable seemed slightly superficial and hence we have revised the statement to the following:

“Engineering the columnar-to-equiaxed transition during rapid solidification in the additive manufacturing process is crucial for its technological advancement. Here, we report a powder size-driven melt pool engineering approach...”

It is indeed true that several different solutions have been implemented to achieve CET in AM. In fact we have listed majority of these solutions in the introduction of our main text both in the first submission as well as in the first revision. On Pg. 2 paragraph 1 of the 1st revision we had stated the following:

“To date, most effective ways to achieve CET in a variety of powder fusion-additive manufacturing (PF-AM) processes include variation of process parameters^{3,12-14}, nanometre scale (nano-scale) inclusion induced heterogeneous nucleation¹⁵, high-intensity ultrasound triggered grain refinement¹⁶ or post-processing driven recrystallization¹⁷. Among these, the last four provide only a unidirectional microstructural control, allowing transitions from columnar to equiaxed but not vice versa. Also, all of these require additional experimental efforts, costs and lack the adaptability to the directed energy deposition (DED) and powder-bed fusion (PBF) PF-AM processes alike. Hence, it is imperative to explore viable alternatives that are economical yet adaptable to achieve either an equiaxed, fine-grained (FG) microstructure or a coarse-grained near monocrystalline microstructure in the as-built condition.”

We do realize that this section did not mention the alloy composition and beam shaping. Hence, we have added these new solutions with discussions in the revised version of the manuscript. In the second revision, the same paragraph reads:

“To date, the most effective ways to achieve CET in a variety of powder fusion-additive manufacturing (PF-AM) processes include variation of processing conditions^{3,12-14}, laser beam shaping¹⁵, alloy composition redesign¹⁶, nanometre scale (nano-scale) inclusion induced heterogeneous nucleation^{16,17}, high-intensity ultrasound triggered grain refinement¹⁸ or post-processing driven recrystallization¹⁹. Among these, the last four have been reported so far in a

capacity of unidirectional microstructure evolution control, allowing transitions from columnar-to-equiaxed and not vice-versa. Also, all of these require additional experimental efforts, costs and lack the adaptability to the directed energy deposition (DED) and powder-bed fusion (PBF) PF-AM processes alike. Hence, it is necessary to explore viable alternatives that are economical yet adaptable to achieve either an equiaxed, fine-grained (FG) microstructure or a coarse-grained near monocrystalline microstructure in the as-built condition.”

It should be noted that parameters such as laser power, scanning speed, scan pattern, and preheating temperature can still be clubbed as processing conditions and hence we did not provide additional references in the revision to main dexterity.

We respectfully disagree with the statement that the state-of-the-art are easier to implement than incorporating different powder sizes. In the revised version we have clearly highlighted to an extent the hardship associated with each of these solutions. We would also like to point out here that before the implementation of solutions such as ultrasonic agitation and beam shaping, other methods existed out there such as 1 – 6 listed by the reviewer. Still the development of these two methods has resulted in furthering the microstructural control achievable in AM even though it requires additional costs to implement the two into the existing AM systems. On the same grounds, we envision our method to be one such extension to the microstructural evolution control. We have however revised the abstract and the main text to rid it of any of the superficial terminology such as that highlighted by the reviewer. We thank the reviewer again for their comments.

Reviewer #3 (Remarks to the Author):

Authors have reported chemical composition analyses. While there are some differences, they are relatively small. This alone does not prove that there are no effects of composition, however the fine powder containing higher interstitials results in coarser grains, suggesting some other effect dominates.

Response: We thank the reviewer for their comments and their previous recommendations for trace element analysis of the two powders. We believe that the minuscule differences observed in the interstitials (C, S, O, N, and H) can be attributed to the environmental effects on the stored powders. In any case, we agree with the reviewer that our powder size driven microstructural control is the dominant effect that results in the generation of coarse grains in fine powders despite them having slightly higher interstitials.